# Rapid modulation of striatal cholinergic interneurons and dopamine release by satellite astrocytes

Jeffrey Stedehouder[1,2,7], Bradley M. Roberts[1,2,7], Shinil Raina [1,2,3], Simon Bossi[1,2,3], Alan King Lun Liu [2,4], Natalie M. Doig[3,5], Kevin McGerty[6], Peter J. Magill [2,3,5], Laura Parkkinen [2,4] & Stephanie J. Cragg [1,2,3] ✉

Astrocytes are increasingly appreciated to possess underestimated and important roles in modulating neuronal circuits. Astrocytes in striatum can regulate dopamine transmission by governing the extracellular tone of axonal neuromodulators, including GABA and adenosine. However, here we reveal that striatal astrocytes occupy a cell type-specific anatomical and functional relationship with cholinergic interneurons (ChIs), through which they rapidly excite ChIs and govern dopamine release via nicotinic acetylcholine receptors on subsecond timescales. We identify that ChI somata are in unexpectedly close proximity to astrocyte somata, in mouse and human, forming a "soma-to-soma" satellite-like configuration not typically observed for other striatal neurons. We find that transient depolarization of astrocytes in mouse striatum reversibly regulates ChI excitability by decreasing extracellular calcium. These findings reveal a privileged satellite astrocyte-interneuron interaction for striatal ChIs operating on subsecond timescales via regulation of extracellular calcium dynamics to shape downstream striatal circuit activity and dopamine signaling.

Striatal dopamine (DA) is crucial for action selection, goal-oriented behavior and reinforcement learning[1]. The release of DA from its extensively branched striatal axonal arbors[2] can be regulated locally by mechanisms that govern axonal excitability, action potential propagation, and release probability[3,4]. A range of striatal neuromodulators and receptors directly and potently modulate striatal DA release in this manner, including, GABA at GABA_{A/B} receptors[4–7], adenosine at A1 receptors[8], and acetylcholine (ACh) released from striatal cholinergic interneurons (ChIs) acting at nicotinic receptors (nAChRs)[9,10]. Consequently, striatal circuits and their regulatory mechanisms are well positioned to govern striatal DA function.

Astrocytes are a major glial cell type in the brain that, besides providing metabolic support, can modulate neuronal signaling[11–13]. Astrocytes tile the brain through territories occupied by their branches and leaflet processes that encompass thousands of neural synapses per astrocyte[14] whose activity they influence by buffering extracellular neuroactive molecules including GABA, glutamate, ATP/adenosine, glycine, K^+, and Ca^{2+} [15]. Release of DA from axons in the striatum has recently been shown to be modulated by striatal astrocytes via their neurotransmitter buffering functions: astrocytic uptake transporters regulate the extracellular levels of GABA and adenosine that inhibit striatal DA release through GABA_{A/B} or A1 receptors respectively[6,8,16]. In parallel, astrocytes

[1]Centre for Cellular and Molecular Neurobiology, Department of Physiology, Anatomy and Genetics, University of Oxford, Oxford OX1 3PT, UK. [2]Oxford Parkinson's Disease Centre, University of Oxford, Oxford OX1 3PT, UK. [3]Aligning Science Across Parkinson's (ASAP) Collaborative Research Network, Chevy Chase, MD 20815, USA. [4]Nuffield Department of Clinical Neurosciences, University of Oxford, Oxford OX3 9DU, UK. [5]Medical Research Council Brain Network Dynamics Unit, Nuffield Department of Clinical Neurosciences, University of Oxford, Oxford OX1 3QT, UK. [6]Mathematical Institute, University of Oxford, Oxford OX2 6GG, UK. [7]These authors contributed equally: Jeffrey Stedehouder, Bradley M. Roberts. ✉e-mail: stephanie.cragg@dpag.ox.ac.uk

in the ventral tegmental area regulate DA neuron activity through regulation of extracellular glutamate and GABAergic interneuron activity[17]. These findings show that astrocytes can modulate striatal DA release via buffering of the extracellular neuromodulator environment.

Astrocytes are however increasingly understood to modulate networks through more than passive buffering[12,13]. Accumulating evidence shows that astrocytes can vary their membrane potential and support subsecond dynamic activity in astrocyte polarization and signaling: They express a broad range of ion channels, transporters and exchangers, including voltage-gated channels[18], whose expression and activity can be regulated by neuronal inputs[19–21] and could influence local membrane potential. Astrocytes can respond to inputs from local neural networks with dynamic intracellular $Ca^{2+}$ signals[22,23] (including in striatum), or with membrane depolarizations, by up to ~20 mV[20,24] on subsecond timescales (-200–300 ms to -1 s) in cortical and hippocampal areas[22,25]. In turn, astrocyte polarization impacts on neural activity e.g. subsecond depolarization of astrocytes inhibits electrogenic glutamate transporter function and increases glutamate neurotransmission onto cortical pyramidal neurons[16]. These dynamic and activity-dependent functions challenge the dogma that astrocytes are merely passive or slow modulators of network activity, and raise the possibility that fluctuating activity in striatal astrocytes might dynamically modulate DA release. Whether and how astrocyte depolarization impacts on DA release has until now remained unknown.

Here, we find that transient optogenetic depolarization of striatal astrocytes potently and rapidly modulates DA release on a subsecond timescale. We reveal that astrocytes locate as soma-to-soma "satellite" astrocytes in preferential close apposition to ChIs and influence ChI activity via rapid modulation of extracellular $Ca^{2+}$ levels, leading to modulation of ACh release, nAChR activation and DA release. These findings reveal a glia-neuron ionic interaction operating in the striatum that governs striatal ChI activity and DA release.

## Results

### Optogenetic depolarization of astrocytes modulates striatal DA release through acetylcholine

To examine the impact of transient depolarization of astrocytes on DA release, we intracranially delivered an AAV encoding a truncated *GFAP* promoter (*GfaABC₁D*, hereafter referred to as 'GFAP')[26] in striatum of adult male and female wildtype mice for astrocyte-specific expression of the light-activated cation channel channelrhodopsin (ChR2-eYFP) three to five weeks later. Viral transduction quantified using a cell-filling variant (GFAP-driven eGFP) was highly specific to S100β-positive astrocytes (-90%) and only very rarely observed in neurons (-0.5%; Supplementary Fig. S1), as reported previously[27–29]. EYFP-labeled astrocytes recorded using whole-cell patch-clamp electrophysiology in ex vivo striatal slices showed typically low input resistance, a linear input current-voltage relationship, and absence of action potentials upon current injection (Fig. 1a–d) as reported previously[11]. Brief blue light pulses (2–500 ms, ~4–5 mW) induced a fast, sub-millisecond onset of a reversible depolarization of ~5–10 mV in ChR2-expressing astrocytes (Fig. 1e) but not in astrocytes from non-ChR2-expressing mice (Supplementary Fig. S2a–e). Depolarization varied with light duration (Supplementary Fig. S2f–j) and, importantly, was equivalent to the level of depolarization seen in astrocytes in a variety of paradigms: in trigeminal main sensory nucleus in response to electrical stimulation of the trigeminal tract afferent inputs[30] or local NMDA application[30]; in hippocampal CA1 in response to input from neuropeptide-Y interneurons[25]; and, furthermore in striatum in response to local application of a striatal neuromodulator, glycine (Supplementary Fig. S2k–o).

We explored the impact of astrocyte depolarization on striatal DA release evoked by a local electrical stimulus and detected with fast-scan cyclic voltammetry (FCV) at carbon-fiber microelectrodes ex vivo in acute dorsolateral striatum (DLS) and nucleus accumbens core

(NAcc) (Fig. 1f, k). In DLS, brief (500 ms) optogenetic stimulation of ChR2-expressing astrocytes but not mCherry-expressing controls reversibly reduced DA release evoked by one electrical pulse (1p, at 400 ms) by ~30% compared to no-light trials (Fig. 1g, h; $F_{(1,23)} = 25.93$, $p < 0.001$, RM ANOVA main effect of the construct). Correspondingly, in GFAP-ChR2 mice there was an increase in the 5p:1p ratio (50 Hz) of DA release (Fig. 1i; $t_{(16)} = 3.741$, $p = 0.002$, paired $t$ test), not seen in mCherry controls (Fig. 1j; $t_{(7)} = 1.948$, $p = 0.092$, paired $t$ test), consistent with a reduction in initial DA release probability. Post hoc data segregation confirmed these astrocyte-induced alterations in DA release were observed in both male and female mice (Supplementary Fig. S3a–c). Similarly, in NAcc, we found that brief (500 ms) astrocyte stimulation reduced DA release evoked by a single electrical pulse (1p) by ~30% (Fig. 1k–m; $F_{(1,13)} = 11.59$, $p = 0.005$, RM ANOVA main effect of the viral construct) and increased the 5p:1p (50 Hz) ratio for DA release in GFAP-ChR2-injected mice (Fig. 1n; $t_{(8)} = 3.194$, $p = 0.013$, paired $t$ test), but not in mCherry controls (Fig. 1o; $t_{(5)} = 1.074$, $p = 0.332$, paired $t$ test). In approximately one third of recording sites in NAcc, we found that light activation of ChR2 alone in the absence of electrical stimulation evoked DA release (Fig. 1l). These data show that brief astrocyte optogenetic depolarization can rapidly modify, and occasionally trigger, DA release throughout striatum.

Striatal astrocytes have recently been shown to regulate tonic inhibition of DA release by $GABA_A$- and $GABA_B$-receptors and adenosine A1 receptors, via astrocytic transporters which limit extracellular GABA[6] and adenosine[8] tone. We tested whether the rapid effects of astrocyte stimulation on electrically evoked DA release involved GABA or A1 receptors. However, the inhibition of DA release seen after astrocyte stimulation involved an alternate mechanism as it was not prevented by the presence of antagonists of $GABA_A$ and $GABA_B$ receptors ((+)-bicuculline; 10 μM, CGP 55845; 4 μM) (Supplementary Fig. S3d, e; $F_{(1,22)} = 0.291$, $p = 0.595$, RM ANOVA main effect of drug) or A1-receptors (8-Cyclopentyl-1,3-dimethylxanthine, CPT; 10 μM) (Supplementary Fig. S3f, g; $F_{(1,24)} = 1.313$, $p = 0.263$, RM ANOVA main effect of drug).

Striatal ACh released from ChIs is an additional potent regulator of DA: activation of ChIs can directly trigger or inhibit DA release (depending on stimulus) through nicotinic ACh receptor (nAChR) activation[9,10,31]. Thus, we next tested a role for ACh and nAChRs. We found that the β2 subunit-containing nAChR antagonist dihydro-β-erythroidine (DHβE; 1 μM) prevented the effects of optogenetic stimulation of astrocytes on DA release evoked by a single electrical pulse (DLS, Fig. 1p; $F_{(1,24)} = 13.67$, $p = 0.001$, RM ANOVA main effect of drug; NAcc, Supplementary Fig. S3h-i; $F_{(1,12)} = 14.30$, $p = 0.003$, RM ANOVA main effect of drug) and on 5p:1p (50 Hz) ratios (Fig. 1q; $t_{(8)} = 2.250$, $p = 0.055$, paired $t$ test).

To identify whether this effect on DA release involved striatal ChIs, and was not due to ACh buffering or release by astrocytes, we tested for occlusion by chemogenetic silencing of ChIs. We virally expressed the *Cre*-dependent inhibitory designer receptor exclusively activated by designer drugs (DREADD)[32] hM4Di-mCherry in adult *ChAT-Cre*[33] mice, and expressed GFAP-ChR2-EYFP in astrocytes as above (Fig. 1r; Supplementary Fig. S4). In *ChAT-Cre*-mCherry control mice (without DREADD construct) in the presence of CNO (1 μM), astrocyte optogenetic depolarization again reduced DA release in DLS by ~25% (Fig. 1s), but in ChAT-hM4Di-mCherry mice this effect was prevented ($F_{(1,8)} = 8.061$, $p = 0.022$, RM ANOVA main effect of DREADD). In parallel, the effect of astrocyte stimulation on 5p:1p (50 Hz) ratio for evoked [DA]$_o$ was occluded (Fig. 1t; mCherry control: $t_{(4)} = 3.238$, $p = 0.032$, paired $t$ test; hM4Di: $t_{(4)} = 1.535$, $p = 0.200$, paired $t$ test). Similarly, application of the broad-spectrum muscarinic ACh receptor agonist oxotremorine-M (oxo-M; 10 μM) which inhibits ChIs[34] and ACh release, also prevented the effect of astrocyte depolarization on evoked DA release in DLS (Supplementary Fig. S3h; $F_{(1,20)} = 12.69$, $p = 0.002$, RM ANOVA main effect of drug), as well as on the 5p:1p

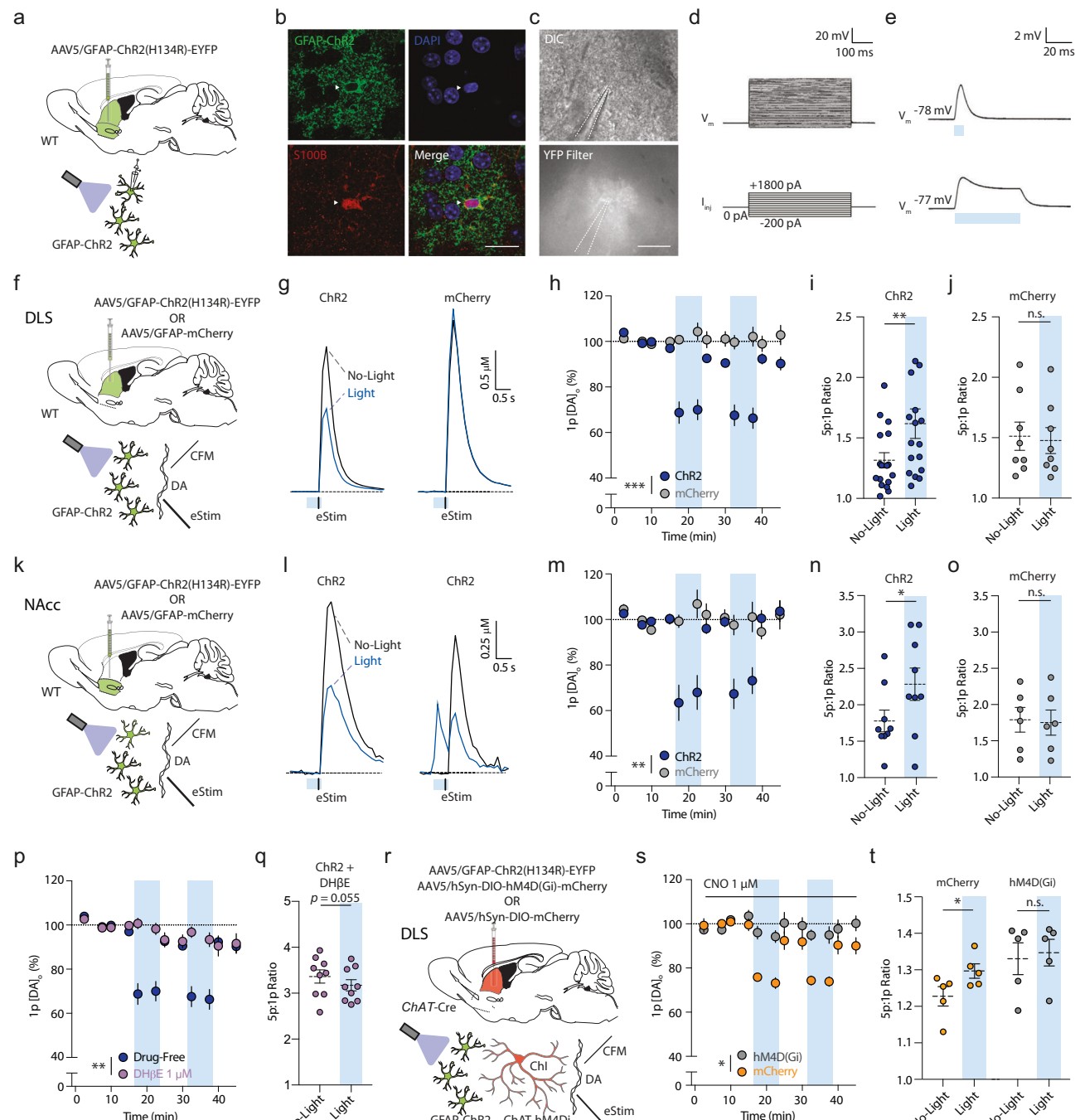

**Fig. 1 | Rapid stimulation of astrocytes modulates striatal DA release via ACh.**
**a, f, k, r** Viral expression, mouse line and recording paradigms. **b** Striatal astrocyte expressing GFAP-ChR2-EYFP (green), DAPI (blue), and S100β-immunoreactivity (red). Scale, 20 μm. **c** *Top*, differential interference contrast (DIC) image and *bottom*, EYFP epifluorescent image, of ChR2-EYFP astrocyte. Scale, 20 μm. **d, e** Example astrocyte membrane voltage ($V_m$) responses to (**d**) current injections ($I_{inj}$) or (**e**) blue light (~4–5 mW) for 5 ms (*top*) or 50 ms (*bottom*). **g** Mean [DA]o transients evoked by one electrical pulse (Estim) without prior light or after light (blue bar, 500 ms) in GFAP-ChR2-expressing (*left*) versus mCherry-expressing controls (*right*). **h, m** Mean peak [DA]o ± SEM evoked by 1p with and without blue light stimulation (blue column) in GFAP-ChR2-expressing (dark blue) versus mCherry controls (gray), normalized to pre-light baseline (dotted line) in (**h**) DLS (ChR2: $n = 17(12)$, mCherry: $n = 8(5)$), or (**m**) NAcc (ChR2: $n = 9(5)$, mCherry: $n = 6(4)$). Trials in which DA release was observed at light onset (**l**, right) were not included in averages. **i–j, n, o** Ratio of peak [DA]o evoked by 5p (50 Hz) versus 1p with and without light activation for ChR2 (**i**, **n**) and mCherry (**j**, **o**) in DLS (**i**, **j**), or NAc (**n**, **o**). ChR2: $n = 17(12)$, mCherry: $n = 8(5)$. **l** *Left*: Mean [DA]o transients evoked in NAc by 1p (Estim) with or without light (blue bar, 500 ms). *Right*: As left, but example light evoking DA release. **p** As (h), but with DHβE (1 μM; gray). Drug-free: $n = 17(12)$, DHβE: $n = 9(6)$. **q** Ratio of peak [DA]o evoked by 5p (50 Hz):1p with and without light-activation with ChR2-expression and DHβE. Drug-free: $n = 17(12)$, DHβE: $n = 9(6)$. **s** Mean peak [DA]o evoked by 1p and (**t**) 5p:1p ratio, with/without blue light (blue column) during CNO application (1 μM) in GFAP-ChR2 mice with expression of hM4Di-mCherry (gray) ($n = 5(3)$) or mCherry controls (yellow) ($n = 5(3)$), normalized to pre-light baseline. ***$p < 0.001$; **$p < 0.01$; *$p < 0.05$. Repeated measures ANOVA in (**h**), (**m**), (**p**), (**s**); Wilcoxon rank sort test in (**q**); paired two-sided $t$ test in (**i**), (**j**), (**n**), (**o**), (**t**). Error bars denote s.e.m. Source data are provided in a Source Data file. Cartoon cell components created in BioRender. Cragg, S. (2023) BioRender.com/k30f956. Cragg, S. (2024) BioRender.com/p68j108.

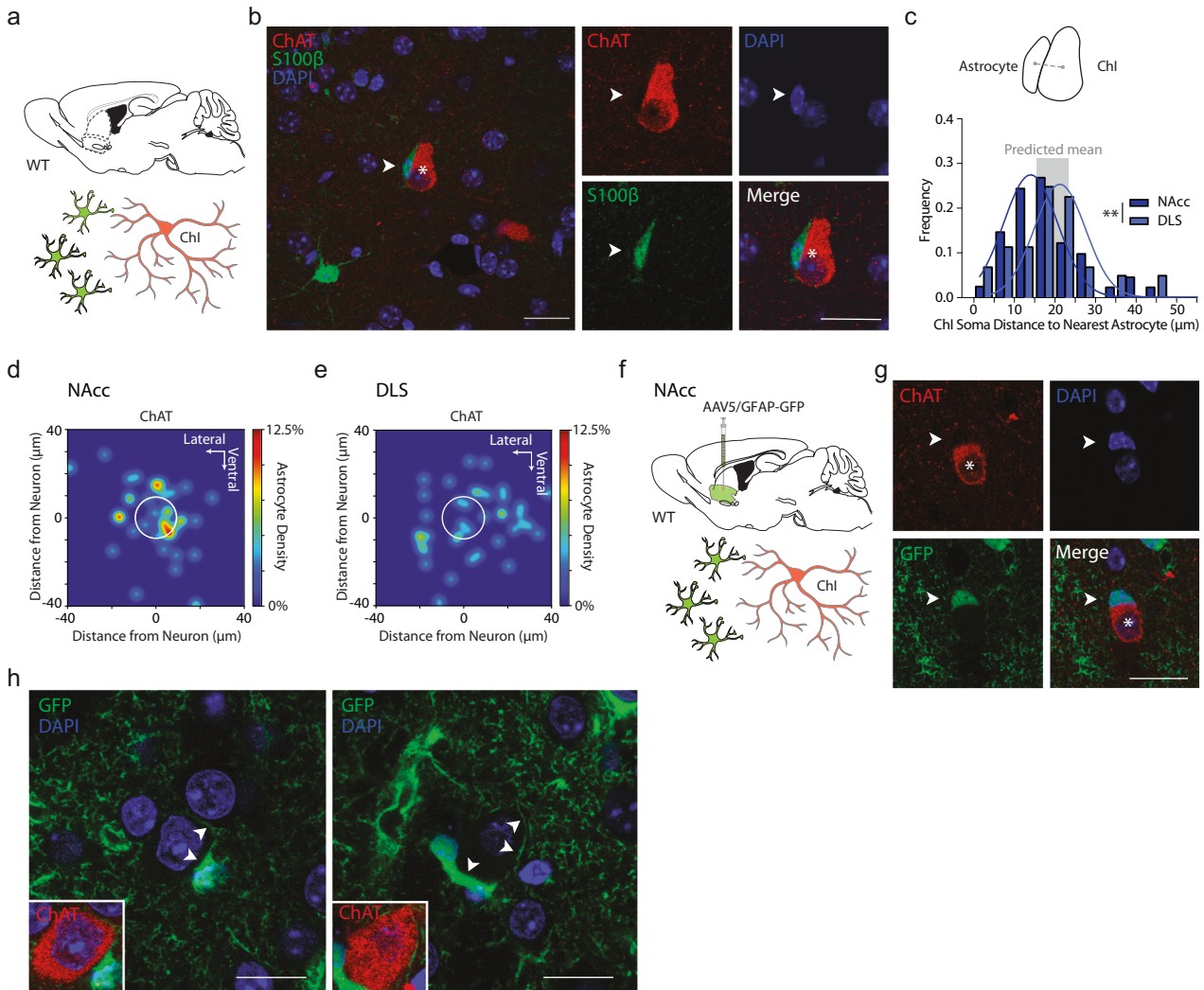

**Fig. 2 | Cholinergic interneurons present soma-to-soma "satellite" astrocytes.**
**a** Experiment schematic: Immunolabelled astrocytes and ChIs in NAcc. **b** *Left*:
Example ChI soma, single-plane confocal image (ChAT; red; asterisk) and accompanying astrocyte (S100β; green; arrowhead), and DAPI+ nuclei (blue). Right:
Single-plane segregated confocal images of ChI soma (asterisk) and closely
accompanying astrocyte (arrowhead). **c** *Top*: Schematic of distance between soma
centers. *Bottom*: Histogram of Euclidian distance between soma centers of neurons
to nearest astrocyte, for ChIs in NAc (blue, $n = 40(4)$) and DLS (gray, $n = 41(4)$),
Gaussian curve-fits. Means: NAc ~16 μm, DLS ~ 19 μm. Gray column, range of predicted means from geometric models. **d**, **e** Heatmaps for location probabilities of
40 astrocytes in relation to nearest ChI in NAc (**d**) or DLS (**e**) (white circle). **f** Viral
expression of GFAP-eGFP in wild-type NAcc and immunolabeling. **g** Example ChI
soma, single-plane confocal image (ChAT; red; asterisk) and accompanying astrocyte (GFP; green; arrowhead), DAPI+ nuclei (blue). **h** Two higher magnification
single-plane confocal examples showing astrocyte processes (GFP; green) surrounding ChI soma (arrowheads). Insert: ChAT immunofluorescence (red).
**p < 0.01. Mann–Whitney *U* test in (**c**). Scale bars, 20 μm. Source data are provided
in a Source Data file. Cartoon cell components created in BioRender. Cragg, S.
(2023) BioRender.com/k30f956. Cragg, S. (2024) BioRender.com/p68j108.

(50 Hz) ratio (Supplementary Fig. S3j, k; $t_{(4)} = 1.472$, $p = 0.215$, paired $t$
test). We also examined directly whether astrocyte activation in NAcc
leads to changes to ACh release from a neuronally-derived source i.e.
one that requires TTX-sensitive neuronal activation. We detected ACh
using the green fluorescent GPCR activation-based (GRAB) sensor for
ACh, GRAB$_{ACh3.0}$[35] in NAcc after activation of red-shifted channelrhodopsin ChrimsonR[36] expressed in astrocytes under a GFAP promotor
(Supplementary Fig. S3l). ChrimsonR-mediated depolarization of NAcc
astrocytes with red light (10 ms, ~4–5 mW) resulted in rapid detection
(<100 ms) of the release of ACh (Supplementary Fig. S3m) that was
prevented by TTX (1 μM; Supplementary Fig. S3m, n; $t_{(4)} = 6.706$,
$p = 0.003$, paired $t$ test) consistent with action-potential-dependent
ACh exocytosis from neurons. Together, these data suggest that brief
optogenetic astrocyte depolarization in DLS and NAcc can rapidly
modulate striatal DA release via a mechanism involving ChIs and
downstream changes to activation of nAChRs on DA axons.

## Striatal ChIs present soma-to-soma "satellite" astrocytes
To understand the mechanism through which astrocytes gate DA via
ChIs, we explored a potential anatomical interaction between astrocytes and ChIs. ChIs are sparse (~1–3% of striatal cells), large, and
spontaneously active cells that provide dense arborization of
striatum[37]. Astrocytes have been noted anecdotally to be found
occasionally in close soma-to-soma proximity to neurons in
striatum[38] but not to spiny projection neurons (SPNs), the predominant population of neurons[39]. "Satellite" perineuronal astrocytes have been observed for pyramidal cells in cortex[29] and
hippocampal CA1[40]. We performed triple immunofluorescent labeling in NAcc and DLS in adult wildtype mice with antibodies to S100β,
DAPI, and choline acetyltransferase (ChAT) or NeuN or parvalbumin
(PV), to assess the relative locations respectively of astrocytes,
nuclei, and ChIs (Fig. 2a–c) or all striatal neurons (Supplementary
Fig. S5f, g), that statistically will be ~97% SPNs (putative SPNs,

pSPNs[11], or PV-positive interneurons as example GABAergic interneurons (Supplementary Fig. S5h).

In both NAcc and DLS, we revealed a close soma-to-soma proximity of a S100β-immunoreactive "satellite" astrocyte to a ChI (Fig. 2b–e): the percentage of examined ChIs for which we observed direct soma-to-soma appositions with an astrocyte membrane (a threshold of more than 3 μm length of apposed cells) was ~43% in NAc (17/40 cells), and ~20% in DLS (8/41 cells). By comparison, for pSPNs, or PV-positive GABA interneurons, the percentage of neurons for which direct appositions to astrocytes were noted in NAc was only ~8% of SPNs (3/38 cells) and ~10% of PV-positive cells (2/20 cells), and in DLS was only ~8% of SPNs (3/39 cells) and ~5% of PV-positive cells (1/20 cells). Mean Euclidian center-to-center distances between a given ChI in NAcc or DLS and their nearest astrocyte were ~16 μm and ~19 μm, respectively (Fig. 2c–e). Mean Euclidean center-to-center distances between a pSPN and nearest astrocytes were not dissimilar, at ~23 μm in NAc and ~21 μm in DLS (Supplementary Fig. S5f,g, not assessed further for PV interneurons). Mathematical modeling (using either a 3D lattice or a uniformly random distribution) predicts that despite their different densities, ChIs and SPNs will have similar mean minimum distances to their nearest neighbor astrocyte (see Supplementary Methods) with a range predicted across models of ~16–23 μm. Since observed soma center-to-center proximities match those predicted distances, and the summed mean radii of typical soma of ChIs (~12.5 μm)[41] plus astrocytes (~6 μm), but not SPNs (~6 μm) plus astrocytes, approach these distances, we propose that the greater proportion of ChIs than SPNs (or PV-interneurons) forming close membrane-to-membrane apposition could be an outcome of the ~two-fold larger cell diameter of ChIs.

We found only a modest inverse correlation between ChI soma size and center-to-center proximity of nearest astrocyte in NAcc (Supplementary Fig. S5a; $r = -0.332$, $p = 0.034$, Pearson correlation), but not in DLS (Supplementary Fig. S5g; $r = -0.100$, $p = 0.534$, Pearson correlation). We found no evidence for associations between astrocyte soma size, ChI-astrocyte proximity, or neighboring ChI soma size in either region (Supplementary Fig. S5).

We validated the satellite-like apposition of astrocytes to ChIs with an additional approach using immunolabeling of ChAT in conjunction with virally mediated GFAP-eGFP expression in astrocytes in wildtype mice (Fig. 2g, h), which better labeled astrocyte soma and processes visibly surrounding ChAT+ somata (Fig. 2h). We confirmed that an antibody against endogenous GFAP labeled less than 10% of striatal astrocytes as shown previously[18] and does not readily reveal satellite astrocytes (not illustrated).

Together, these findings reveal that ChIs have an unexpectedly close spatial proximity to S100β-positive striatal astrocytes in a satellite-like soma-to-soma apposition in DLS and NAcc.

### Striatal astrocytes can rapidly modulate ChI spontaneous activity

We next examined the physiological influence of astrocyte depolarization on ChIs. We virally co-expressed and visualized a Cre-dependent mCherry (DIO-mCherry) in striatal ChIs in adult *ChAT-Cre* mice, and expressed GFAP-ChR2-EYFP in striatal astrocytes in NAcc as described above (Fig. 3a). After ~3–5 weeks, we performed ex vivo targeted whole-cell electrophysiological recordings of ChAT:mCherry + cells (Fig. 3b) within the area of GFAP-ChR2-EYFP co-expression in NAcc. Striatal mCherry-positive neurons showed relatively depolarized membrane potentials between action potentials, large $I_h$, and broad action potentials with a large afterhyperpolarization highly characteristic of ChIs (Fig. 3c; Supplementary Table S1). Current clamp recordings showed a regular (39/41 cells; 95.1%) or occasionally a bursting (2/41 cells, 4.9%) spontaneous firing pattern (Fig. 3d), as reported previously in vivo[42] and ex vivo[43].

A brief blue light pulse (2–500 ms; ~4–5 mW) to depolarize astrocytes induced rapid membrane potential changes in ~80% of recorded ChIs, with concomitant changes in firing rate (33/41 cells, Fig. 3e-j). ChIs from control *ChAT-Cre* mice expressing *Cre*-dependent mCherry and GFAP-driven GFP without ChR2 did not show such responses to light (Supplementary Fig. S6a-c). Intriguingly, ChI initial responses to optogenetic depolarization of astrocytes were bidirectional and followed a bimodal distribution: some cells initially increased firing, dubbed 'excited' cells (12/41; 29.3%; $t_{(11)} = 7.933$, $p < 0.001$; paired $t$ test), while others showed only a decrease in firing, dubbed 'inhibited' cells (21/41; 51.2%; Fig. 3e-i; $t_{(20)} = 10.650$, $p < 0.001$; paired $t$ test). The remainder of cells were classified as non-responders (8 out of 41; 19.5%; $t_{(7)} = 1.678$, $p = 0.137$; paired $t$ test). Cells did not change their response direction during successive trials (Supplementary Fig. S6d, e). The onset times of excitatory responses occurred within ~1–10 ms of light onset, whereas inhibitory responses occurred with significantly longer latency, after ~5–70 ms (Fig. 3j; $U = 5.000$, $p < 0.001$, Mann–Whitney $U$ test) suggesting distinct mechanisms. Excitatory and inhibitory firing rate responses both scaled with light duration (Supplementary Fig. S6f, g) and were observed in mice of both sexes (Supplementary Fig. S6h, i). Bidirectional responses were not restricted to transgenic *ChAT-Cre*::mCherry-expressing mice, but could also be observed in wildtype animals expressing GFAP-ChR2-EYFP, in putative ChIs (pChIs) which were filled with biocytin and identified *post hoc* with ChAT immunofluorescence (Supplementary Fig. S6j–n).

Neither excitatory or inhibitory response was consistent with direct light activation of ChIs through non-selective expression of GFAP-ChR2 in ChIs. Direct ChR2 expression and activation of ChIs with light showed faster responses yet (<1 ms onset) followed by lower trial-to-trial jitter in the first elicited action potential compared to light activation of GFAP-ChR2-EYFP (Supplementary Fig. S7a–f). Furthermore, we were unable to detect aberrant ChR2-EYFP expression in recorded ChIs when expressing GFAP-ChR2-EYFP ($n = 10(3)$). We filled a set of ChIs with biocytin for *post hoc* immunofluorescent labeling and amplified GFAP-ChR2-EYFP with anti-GFP (see Methods) for detailed morphological scrutiny of potential overlap under confocal microscopy. We could not, however, detect colocalization of amplified EYFP with recorded neurons in somata of any examined cells (Supplementary Fig. S7g, h), but could instead corroborate the close apposition of astrocyte processes to the ChI somata (as in Fig. 2). Together, these data argue against the effects of GFAP-ChR2 arising from aberrant expression in ChIs.

After light offset, the majority of ChIs (28/41; 68%) showed a consecutive brief (~500 ms) period of decreased spiking activity, followed by a normalization of firing rate back to baseline rates after ~500–1000 ms (Fig. 3i, Supplementary Fig. S8a, b), reminiscent of the well-described ChI pause[44]. These secondary inhibition effects at light offset had a unimodal distribution of responses, and z-score normalized firing rates during light on and offset periods were uncorrelated (Supplementary Fig. S8c, d), suggesting uncoupled mechanisms for onset/offset effects.

We explored whether the initial excitation or inhibition of ChIs to astrocyte depolarization was a consequence of a direct interaction between astrocytes and ChI, or whether they depended on activity in other local circuit neurons, by assessing dependence on voltage-gated Na+-channels required for action potential generation. In ChIs responding to astrocyte activation with excitation, the underlying depolarization of ChI membrane potential was reduced only ~8% by voltage-gated Na+-channel blocker tetrodotoxin (TTX, 1 μM; Fig. 3k; $t_{(8)} = 2.564$, $p = 0.034$, paired $t$ test), suggesting a minimal role for action potentials in other circuits, and instead indicating a direct depolarizing mechanism. In ChIs responding with inhibition, the underlying hyperpolarization of membrane potential was prevented by TTX (1 μM; Fig. 3k; $t_{(10)} = 5.647$, $p < 0.001$, paired $t$ test) indicating a

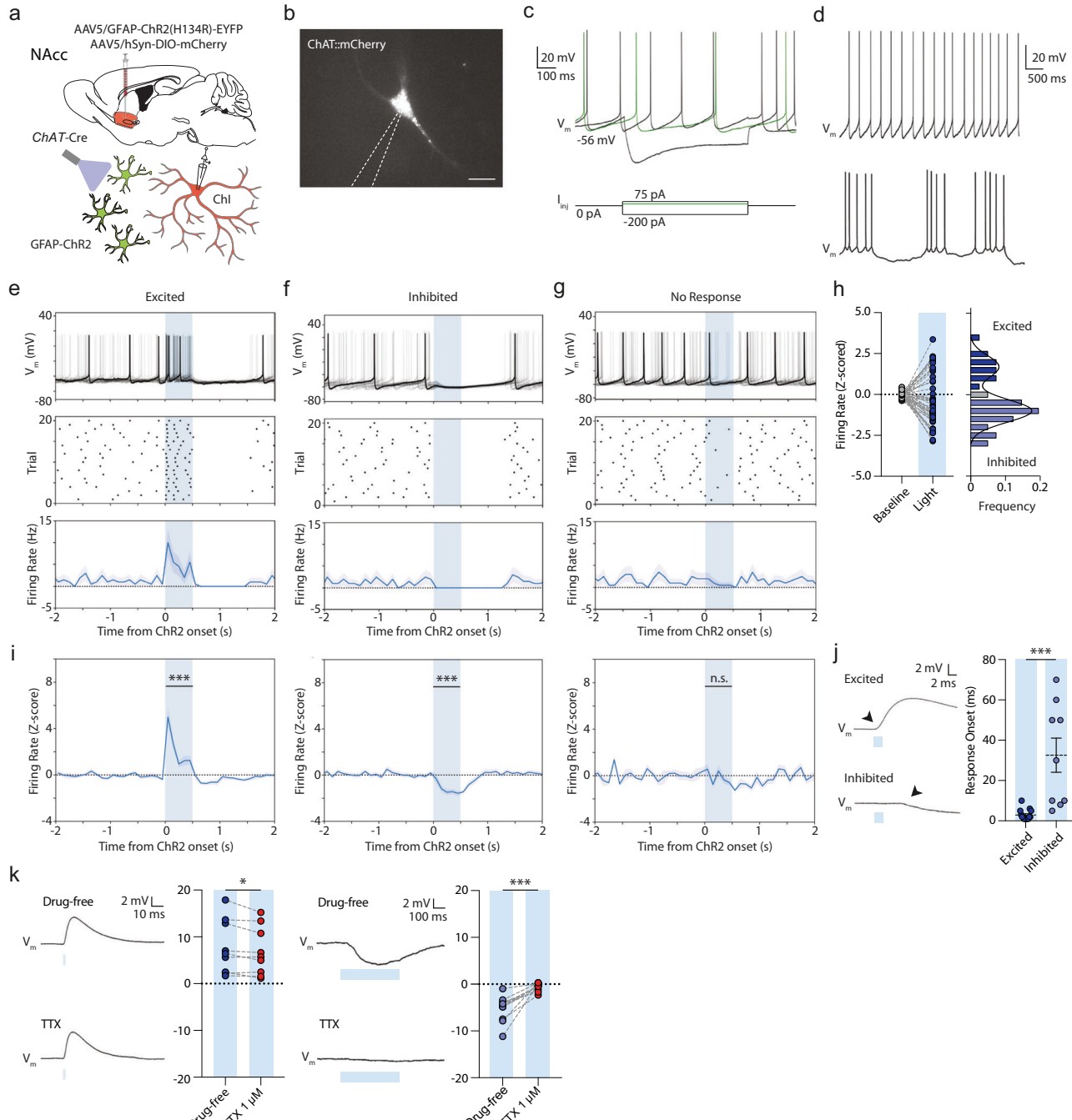

**Fig. 3 | Astrocytes rapidly modulate ChI excitability and spontaneous firing.**
**a** Schematics for viral expression, mouse lines and recording paradigms. Cartoon cell components created in BioRender. Cragg, S. (2023) BioRender.com/k30f956. Cragg, S. (2024) BioRender.com/p68j108. **b** Targeted whole-cell recording of mCherry-expressing ChI under DsRed epifluorescence. Scale bar, 20 μm. **c, d** Examples of membrane potential responses of ChIs to current injections (**c**) and rhythmic spontaneous activity at 0 pA holding potentials (*top*), or occasionally, bursting activity (*bottom*) (**d**). **e–g** Example membrane voltage traces (*top*), action potential raster plot (*middle*), and spontaneous firing frequency (100 ms bins; *bottom*) for 20 consecutive trials showing ChIs that (**e**) were excited, (**f**) inhibited (**g**) or unresponsive to blue light activation of ChR2-expressing astrocytes (500 ms, blue column).

Shaded areas denote s.e.m. **h** *Left*: Z-scored firing rates during baseline (1000 ms before light) and light ($n = 41(13)$). *Right*: Bimodal distribution of Z-scored responses. **i** Z-scored firing rates grouped by response (*left*: $t_{(11)} = 7.933$, $p < 0.001$; paired $t$ test; *mid*: $t_{(20)} = 10.650$, $p < 0.001$; paired $t$ test; *right*: $t_{(7)} = 1.678$, $p = 0.137$; paired $t$ test). **j** *Left*: Membrane voltage responses to 2 ms blue light for non-spontaneously active cells. *Right*: Response onset times. (Excited: $n = 15(11)$; Inhibited: $n = 9(6)$. $U = 5.000$, $p < 0.001$; Mann–Whitney $U$ test. **k** Membrane voltage responses to blue light activation for non-spontaneously active cells in drug-free conditions or in presence of TTX (1 μM) for excited (*left*: $n = 9(6)$) and inhibited cells (*right*: $n = 11(8)$. ***$p < 0.001$; **$p < 0.01$; *$p < 0.05$. Paired two-sided $t$ test in (**i**), (**k**); Mann–Whitney $U$ test in (**j**).). Error bars denote s.e.m. Source data are provided in a Source Data file.

requirement for action potential generation in other local circuit neurons and an indirect mechanism. Thus, ChI excitation is the most rapid and direct effect of astrocyte activation, while inhibition with longer latency is an indirect, downstream outcome.

We compared the effects of astrocyte stimulation on ChIs to the responses of striatal SPNs. In whole-cell electrophysiological recordings in NAcc, putative SPNs (pSPNs) showed characteristic hyperpolarized resting membrane potentials, low input resistance, and latency

to first action potential on current injection (Supplementary Fig. S9a, b). In response to optogenetic depolarization of GFAP-ChR2-EYFP astrocytes, ~90% of pSPNs recorded in NAcc (26/29 cells; 89.6%) showed unimodal small depolarizations (~4 mV) but not action potentials (0/29 cells; 0%; Supplementary Fig. S9c–h). These depolarizations were blocked by TTX (1 μM; Supplementary Fig. S9i–k), indicating an indirect effect, in contrast to the depolarizations observed in ChIs. Together, these data suggest that brief astrocyte depolarization can act indirectly via modulation of local neural circuits to hyperpolarize ChIs and excite SPNs, but most directly, it can depolarize ChIs through a rapid mechanism, consistent with a privileged satellite-like proximity between astrocytes and a population of ChIs.

### Excitation of ChIs by astrocytes is independent of gap junctions or fast ionotropic receptor transmission

How can astrocytes rapidly and directly excite ChIs? At fast timescales, and given the satellite-like close proximity (Fig. 2), potential candidate mechanisms are electrical coupling between astrocytes and ChIs through gap junctions, ionotropic glutamatergic or GABAergic transmission via postulated glial release, glial ATP/adenosine release, or ionic mechanisms (e.g. for $K^+$, $Ca^{2+}$, $Cl^-$).

Application of the gap junction blocker carbenoxolone (CBX; 100 μM) failed to prevent excitatory responses (Supplementary Fig. S11a, b, g; $t_{(8)} = 1.573$, $p = 0.154$, paired $t$ test). In addition, biocytin filling of targeted ChIs did not reveal spread of labeling to any adjacent cells (Supplementary Fig. S10a–c). Conversely, biocytin filling of recorded astrocytes labeled a large number of local other astrocytes suggesting an intact astrocytic syncytium, but no biocytin was observed in any NeuN+ neurons (Supplementary Fig. S10d–f). These data indicate that gap junctions do not mediate astrocyte modulation of ChIs. Light-induced excitation of ChIs was furthermore not prevented by application of: ionotropic glutamate NMDA and AMPA receptor blockers AP-V (50 μM) and NBQX (5 μM), (Supplementary Fig. S11c, g; $t_{(5)} = 1.441$, $p = 0.209$, paired $t$ test); GABA_A receptor antagonist (+)-bicuculline (10 μM; Supplementary Fig. S11d, g; $t_{(6)} = 0.026$, $p = 0.980$, paired $t$ test); adenosine $A_1/A_{2A}$ receptor antagonist caffeine (20 μM) (Supplementary Fig. S11e, g; $t_{(6)} = 1.448$, $p = 0.198$, paired $t$ test); or, purinergic P2 receptor antagonist suramin (100 μM) (Supplementary Fig. S11f, g; $t_{(5)} = 0.216$, $p = 0.838$, paired $t$ test). Astrocytic excitation of ChIs is therefore independent of gap junction coupling, ionotropic glutamatergic or GABAergic transmission, or astrocytic modulation of ATP or adenosine, and thus, an alternative mechanism is involved.

### Inhibition of ChIs by astrocytes occurs through polysynaptic GABAergic inhibition

Before exploring the mechanism further, we compared how astrocytes act indirectly through TTX-sensitive mechanisms to inhibit ChIs and excite SPNs (Fig. 3, S6i–k). ChI inhibition was not prevented by gap junction blocker carbenoxolone (CBX; 100 μM) (Supplementary Fig. S11h, i, m; $t_{(7)} = 0.938$, $p = 0.380$, paired $t$ test), DA D2 receptor antagonist L-741626 (1 μM) (Supplementary Fig. S11k, m; $t_{(4)} = 0.361$, $p = 0.736$, paired $t$ test), or nAChR antagonist DHβE (1 μM) (Supplementary Fig. S11l, m; $t_{(8)} = 0.804$, $p = 0.444$, paired $t$ test), indicating that inhibition of ChIs was not downstream of the release of DA[44] or ACh acting at nAChRs[45,46], However, it was attenuated by the GABA_A receptor antagonist bicuculline (10 μM; Supplementary Fig. S11j, m; $t_{(9)} = 7.318$, $p < 0.001$, paired $t$ test) which, together with its TTX-sensitivity (Fig. 3), indicates a synaptically-mediated GABAergic inhibition. Potential sources of GABA besides GABA neurons could include the neuronal population of ChIs that are excited by astrocytes, as ChIs have been shown to co-release GABA[45,47].

The TTX-sensitive depolarization of pSPNs by astrocytes (see Supplementary Fig. 9) also required GABA_A receptors. Light-induced fast depolarizing responses in pSPNs were blocked by the application of the GABA_A receptor antagonist (+)-bicuculline (10 μM; Supplementary Fig. S9l–n; $t_{(11)} = 3.178$, $p = 0.009$, paired $t$ test) (but a slow residual component remained unaffected by both TTX and bicuculline) (Supplementary Fig. S9p, q). These data indicate that rapid astrocytic modulation of pSPNs occurs indirectly via a neuronal source of GABA release. These underlying circuits were not the focus of interest here and were not explored further.

### Astrocytes rapidly excite ChIs through local clearance of $[Ca^{2+}]_o$

To identify the mechanism mediating the rapid excitation of ChIs by astrocytes, we next considered ionic mechanisms. Astrocytes express a broad range of ion channels, transporters and exchangers to modulate intracellular and extracellular concentrations[11] of ions including $[K^+]_o$ and $[Ca^{2+}]_o$[30,48,49] to which ChIs are particularly sensitive[50].

We first examined whether the excitation versus inhibition response of ChIs to astrocyte optogenetic depolarization was correlated with intrinsic ChI membrane properties or action potential characteristics in the absence of astrocyte depolarization, as these intrinsic properties can be influenced by the ionic environment. We found that the response direction of ChIs to astrocyte depolarization were correlated with subtle intrinsic electrophysiological differences in the absence of stimulation (Fig. 4). Response groups did not differ in resting membrane potential, action potential threshold, action potential amplitude, input resistance, or action potential half-width, but ChIs that were excited by astrocyte depolarization displayed a slightly smaller (~5%) voltage sag during hyperpolarizing (negative) current injections (Fig. 4a,e; $t_{(59)} = 3.963$, $p < 0.001$, unpaired $t$ test), larger spike frequency adaptation in response to suprathreshold current injections ($U = 145$, $p < 0.001$, Mann–Whitney $U$ test) as well as smaller (~25%) after-hyperpolarization amplitudes following a single action potential compared to ChIs that were inhibited by astrocytes (Fig. 4i, k; $t_{(62)} = 3.376$, $p = 0.001$, unpaired $t$ test). These findings raise the possibilities that either the outcome of astrocyte depolarization on ChIs was governed by intrinsic ChI properties, or vice versa that these properties of ChIs were established by the influence of closely appositioned astrocytes.

These divergent intrinsic properties have been reported to be $[Ca^{2+}]_o$ or $[K^+]$-dependent[43,50,51] and we validated that these intrinsic properties in wildtype ChIs were sensitive to $[Ca^{2+}]_o$ in a bidirectional manner: Decreasing or increasing $[Ca^{2+}]_o$ in the artificial cerebral spinal fluid (aCSF) respectively lowered or raised each of the voltage sag, the medium afterhyperpolarization (mAHP) amplitude, and spike frequency adaptation (Fig. 4j–l; sag: $F_{(2,30)} = 4.373$, $p = 0.022$; mAHP amplitude: $F_{(2,30)} = 51.560$, $p < 0.001$; adaptation: $F_{(2,30)} = 10.060$, $p < 0.001$, one-way ANOVA). Manipulations of $[K^+]_o$, by contrast, led to more complex modulation of membrane characteristics that did not vary bidirectionally with increasing versus decreasing $[K^+]_o$ (Supplementary Fig. S12). These findings show that divergent intrinsic properties of ChIs that are associated with excitation or inhibition responses to astrocyte activation can be induced bidirectionally by alterations to $[Ca^{2+}]_o$ but not $[K^+]_o$. While $[K^+]_o$ could continue to be a contributing factor, these findings raise the possibilities that local differences in $[Ca^{2+}]_o$ might occur and confer physiological properties on different neurons and moreover, that astrocytes might modulate ChIs via $[Ca^{2+}]_o$.

We tested the hypothesis that striatal astrocytes influence ChI activity by modulating $[Ca^{2+}]_o$ under optogenetic activation. We fabricated $Ca^{2+}$-selective microelectrodes and validated their selective sensitivity to $[Ca^{2+}]$, but not $[Mg^{2+}]$, $[Zn^{2+}]$, or $[K^+]$ during calibrations in vitro (Supplementary Fig. S13a–c). In ex vivo acute striatal slices from wildtype mice, $Ca^{2+}$-selective microelectrodes showed rapid lowering of $[Ca^{2+}]_o$ after local puff application of the calcium chelators EGTA (5 mM) or BAPTA (5 mM), or elevation of $[Ca^{2+}]_o$ after puff application of a high $Ca^{2+}$ aCSF (20 mM) (Supplementary Fig. S13d, e) with subsecond resolution. Upon astrocyte optogenetic stimulation in NAcc, we detected rapid and transient lowering of

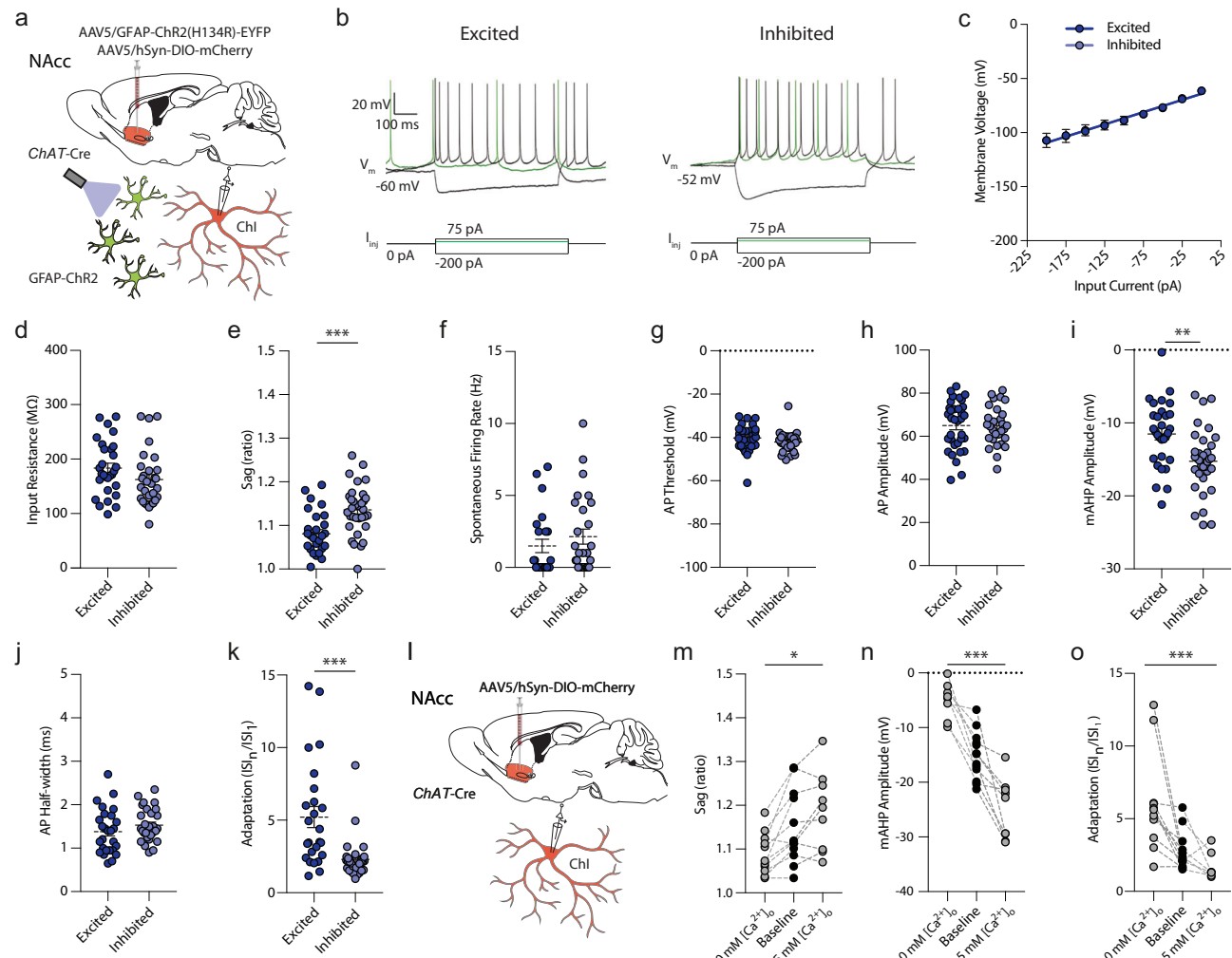

**Fig. 4 | Differential electrophysiological properties of ChIs and the influence of $[Ca^{2+}]_o$. a, l** Schematics for viral expression, mouse lines and recording paradigms. **b)** Electrophysiological current-clamp recordings of mCherry-expressing ChIs in response to 750-ms square negative, positive, and threshold (green) current injections for cells excited (*left*) or inhibited (*right*) by stimulation of ChR2-expressing astrocytes. Cells show rhythmic spontaneous activity, depolarized membrane potential, voltage sag in response to negative current injections, and a large afterhyperpolarization (AHP). **c** *I–V* relationships. Excitation: $n = 32(15)$; inhibition: $n = 30(14)$. **d–k** Excited ChIs compared to inhibited ChIs showed (**d**) a trend to higher input resistance compared to inhibited ChIs ($U = 316$, $p = 0.072$, Mann–Whitney $U$ test), (**e**) a smaller sag ratio ($t_{(59)} = 3.963$, $p < 0.001$, unpaired $t$ test), (**f**) no differences in spontaneous action potential activity ($U = 266$, $p = 0.273$, Mann–Whitney $U$ test), (**g**) AP threshold ($t_{(58)} = 1.505$, $p = 0.138$, unpaired $t$ test)), or (**h**) AP amplitude ($t_{(51)} = 0.348$, $p = 0.729$, unpaired $t$ test), (**i**) higher mAHP voltage

($t_{(62)} = 3.376$, $p = 0.001$, unpaired $t$ test), (**j**), no difference in AP half-width ($t_{(54)} = 1.269$, $p = 0.210$, unpaired $t$ test), and (**k**) larger spike frequency adaptation ($U = 145$, $p < 0.001$, Mann–Whitney $U$ test) from the first interspike interval ($ISI_1$) to the last ($ISI_n$) during a positive 750-ms square current pulse at +100 pA. Excitation: $n = 32(15)$; inhibition: $n = 30(14)$. **m–o** Lowering extracellular aCSF $[Ca^{2+}]_o$ from baseline (1.6 mM) to 0 mM decreases the sag ratio (m; $F_{(2,30)} = 4.373$, $p = 0.022$, one-way ANOVA), raises the voltage of the mAHP (n; $F_{(2,30)} = 51.56$, $p < 0.001$, one-way ANOVA), and increases spike frequency adaptation (o; $F_{(2,30)} = 10.06$, $p < 0.001$, one-way ANOVA). Excitation: $n = 32(15)$; inhibition: $n = 30(14)$. Increasing extracellular aCSF $[Ca^{2+}]_o$ to 5 mM has opposite effects. ***$p < 0.001$; **$p < 0.01$; *$p < 0.05$. Mann–Whitney $U$ test in (**k**), unpaired two-sided $t$ test in (**e**) and (**i**). One-way ANOVA, (m-o). Error bars denote s.e.m. Source data are provided in a Source Data file. Cartoon cell components created in BioRender. Cragg, S. (2023) BioRender.com/k30f956. Cragg, S. (2024) BioRender.com/p68j108.

$[Ca^{2+}]_o$ by ~100 μM (Fig. 5a, b; $t_{(31)} = 3.349$, $p = 0.002$, unpaired $t$ test), a value that likely underrepresents changes occurring closer to a source mechanism or small intercellular space. These effects on $[Ca^{2+}]_o$ were insensitive to TTX (1 μM; Fig. 5c; $t_{(3)} = 0.087$, $p = 0.937$, paired $t$ test), and therefore were not a result of action potential activity in the neuronal network.

If astrocyte depolarization promotes ChI activity through this reduction in $[Ca^{2+}]_o$, then manipulations to $[Ca^{2+}]_o$ should accordingly modulate or even mimic the impact of astrocyte depolarization. Indeed, we found that we could partly overcome the modulation of ChI activity induced by astrocyte optogenetic stimulation in the presence of elevated $[Ca^{2+}]_o$ (5 mM vs 1.6 mM) (Fig. 5d, e; $t_{(4)} = 3.564$, $p = 0.024$, paired $t$ test), and could mimic excitation by brief local somatic

application of a calcium chelator (Fig. 5f–h): Puff application of EGTA or BAPTA (5 mM; ~500 ms) rapidly and reversibly enhanced spontaneous firing rates in all examined ChIs engaged in regular single-spike spontaneous firing (Fig. 5f–h; EGTA: 15/15; 100%; $t_{(14)} = 9.268$, $p < 0.001$, paired $t$ test; BAPTA: 12/12; 100%; $t_{(11)} = 9.506$, $p < 0.001$, paired $t$ test; aCSF: $t_{(5)} = 1.240$, $p = 0.270$, paired $t$ test) or bursting (($n = 3(2)$); Supplementary Fig. S13f, g). These data show that brief local $[Ca^{2+}]_o$ reductions are sufficient for rapid and reversible increases in firing of ChIs, and that striatal astrocytes rapidly and reversibly modulate ChI excitability through dynamically regulating local $[Ca^{2+}]_o$.

The mechanisms through which a decrease in $[Ca^{2+}]_o$ is coupled to enhanced ChI excitability might be numerous, given the expression of SK[50], BK[52], HCN[50], and potentially $Ca^{2+}$-screened NALCN channels in

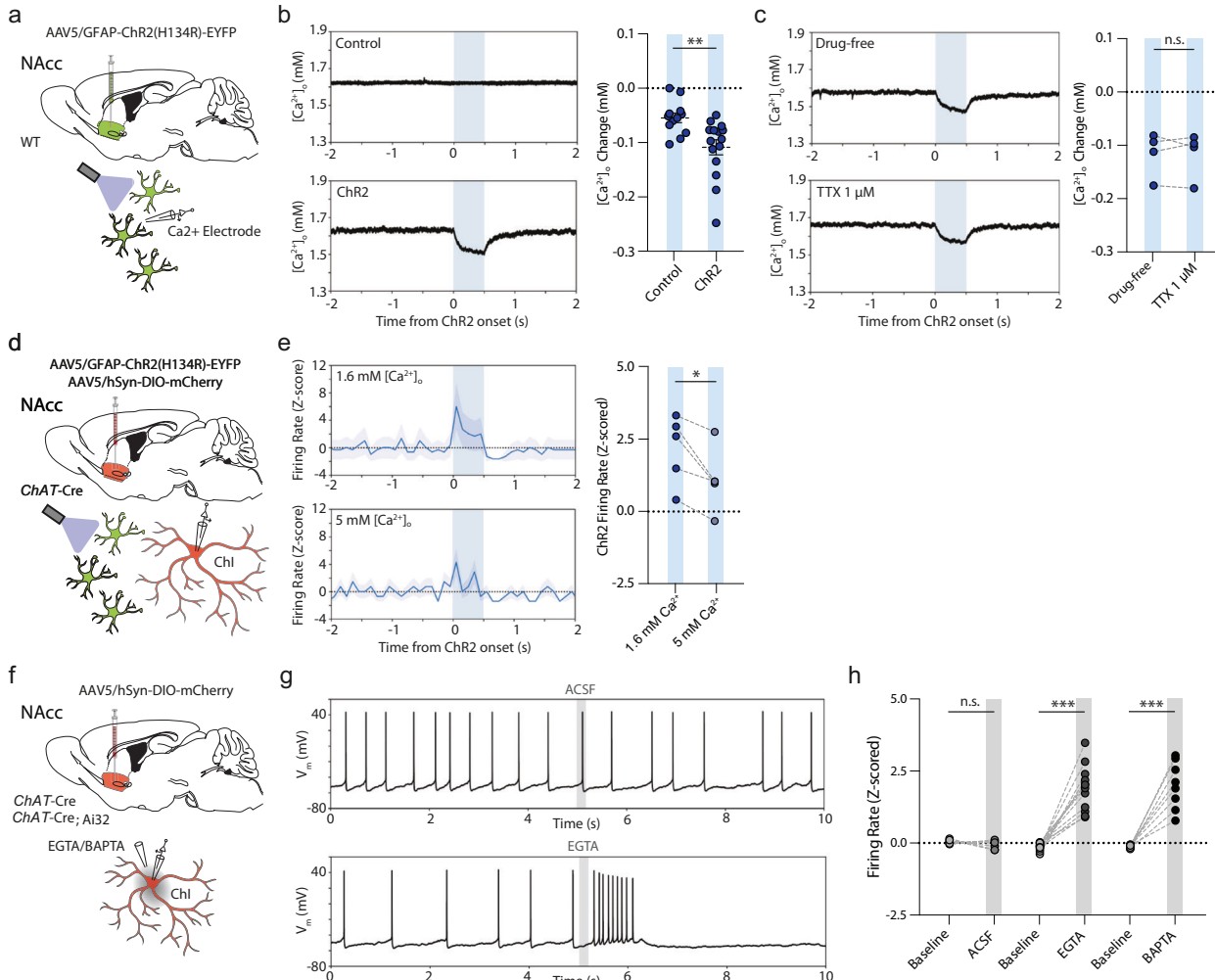

**Fig. 5 | Astrocytes excite ChIs through rapid decrease in $[Ca^{2+}]_o$. a, d, f** Schematics for viral expression, mouse lines and recording paradigms. **b, c** *Left,* example traces; *right,* changes in $[Ca^{2+}]_o$ in response to 500-ms blue light activation (blue column) in (**b**) controls (*n* = 15(5)) and GFAP-ChR2 mice (*n* = 18(5)) and (**c**) GFAP-ChR2 mice in drug-free conditions or with TTX (1 μM), *n* = 4(4). **e** *Left,* z-scored firing rate responses, and *right,* z-scored mean firing rate changes for excited ChIs in under light-activation (blue bar) in 1.6 mM or 5 mM $[Ca^{2+}]_o$. *n* = 5(4). Shaded areas denote s.e.m. **g** Raw membrane voltage responses of recorded ChIs with puff application of aCSF (*top*) or EGTA (*bottom*) at 5 s (gray bar). **h** Z-scored firing rate changes for aCSF (*n* = 6(5)), calcium chelator EGTA (5 mM) or BAPTA (5 mM) (EGTA: *n* = 12(6); BAPTA: *n* = 9(5)) compared to baseline. ****p* < 0.001; ***p* < 0.01; **p* < 0.05; Paired two-sided *t* test in (**c**), (**e**), (**h**). Unpaired *t* test in (**b**). Error bars denote s.e.m. Source data are provided in a Source Data file. Cartoon cell components created in BioRender. Cragg, S. (2023) BioRender.com/k30f956. Cragg, S. (2024) BioRender.com/p68j108.

ChIs, and were not pursued here. However, we note that the medium-duration afterhyperpolarization (mAHP) that follows ChI spikes and is supported by SK channels[52] was smaller in the presence of EGTA (Fig. 5g, Supplementary Fig. S14), and after astrocyte activation (Fig. 3e, Supplementary Fig. S14), in line with ChI activation via at least reduced SK activation.

### Satellite astrocytes can accompany striatal ChIs in human post-mortem brain

Finally, to understand the translational value to humans of this interaction between astrocytes and ChIs in mouse, we examined human anterior striatum for a similar configuration between striatal astrocytes and ChIs. We performed double immunofluorescent staining formalin-fixed paraffin-embedded post-mortem brain sections from three human healthy control subjects (Supplementary Table S3) for ChIs and astrocytes using antibodies for ChAT and GFAP (Fig. 6a), and anecdotally found that ChIs in human striatum are similarly frequently accompanied by GFAP-immunopositive satellite astrocyte somata and processes in putamen and NAcc (Fig. 6b–d).

## Discussion

We reveal a distinctive mode of astrocyte-interneuron communication in the striatum through which astrocytes act as active components regulating striatal neural circuits on a rapid timescale. We find that astrocytes in the mouse and human striatum occupy a privileged satellite proximity to ChIs. Transient astrocyte depolarization, through local modulation of extracellular calcium, directly and rapidly excites a population of ChIs at subsecond timescales, with downstream effects on other ChIs, wider striatal neuronal networks, and a strong impact on DA release. These findings identify ACh as a neuromodulator governed directly by astrocytes, and as one mediating subsecond astrocyte-DA axon modulation. They add to a growing body of data indicating that astrocyte-neuron interactions shape DA transmission, and at subsecond timescales more rapid than previously appreciated.

### Satellite astrocytes to striatal ChIs

We found a potential anatomical basis for a targeted and rapid interaction between astrocytes and striatal ChIs. Using a dual immunofluorescent approach in wildtype mice and in human brains, we found

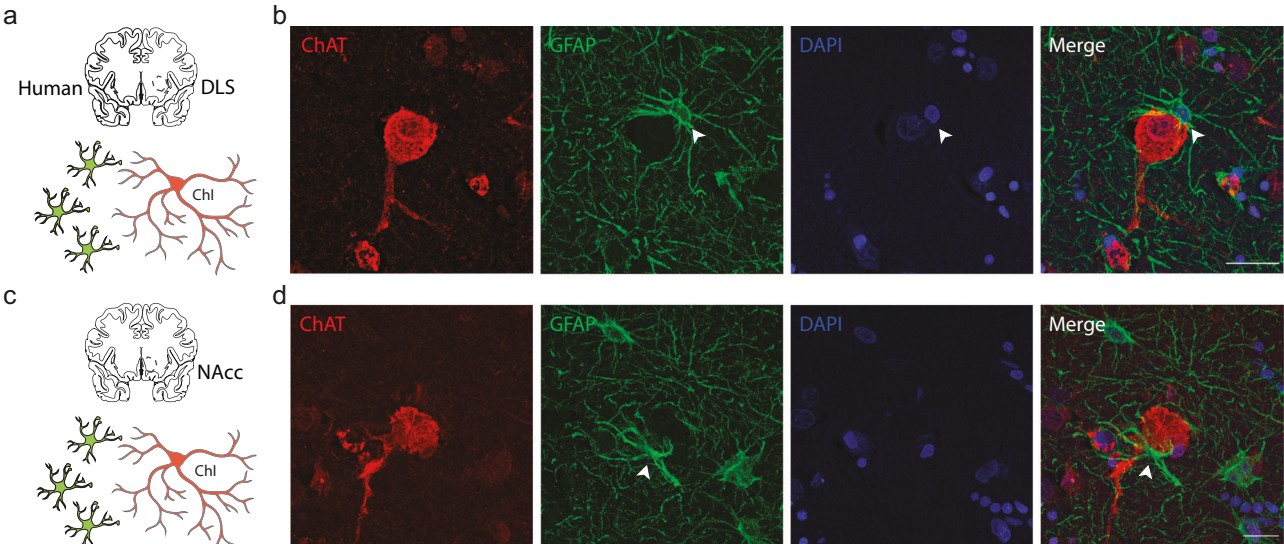

**Fig. 6 | Examples of human ChIs with soma-soma satellite astrocytes.**
**a, c** Experiment schematics. **b, d** Maximum projection confocal images showing human ChIs (ChAT; red), closely accompanying and ensheathing astrocyte (GFAP; green; arrowhead), and DAPI+ nuclei (blue) in (**b**) human putamen and (**d**) human NAcc. Scale bar, 20 μm. Source data are provided in a Source Data file. Cartoon cell components created in BioRender. Cragg, S. (2023) BioRender.com/k30f956. Cragg, S. (2024) BioRender.com/p68j108.

that ChIs, but not striatal neurons generally, are located in close membrane-membrane proximity to astrocytes in a soma-to-soma satellite configuration. In mouse DLS, neurons outnumber astrocytes by ~6 to 1, whereas this number is closer to parity in hippocampus[18,39], and the territory enclosed by an entire astrocyte volume on average intersects with ~16–20 neuronal soma in DLS[18,39], a much higher number than in hippocampal CA1 (~1–13 neurons depending on subregion)[18,53], cortex (~6 neurons)[54] or amygdala (~5 neurons)[53], and through a territory volume larger than occupied for hippocampal astrocytes[18]. ChIs only represent ~2% of the striatal neuronal population[55], and will be outnumbered by astrocytes by ~8 to 1, potentially offering ample opportunity for astrocytes to occupy a privileged soma-soma topography for ChIs. Our empirical observations and mathematical modeling indicate however that for average ChI or SPNs, their mean nearest neighbor astrocyte is at a similar, and predictable, soma center-soma center distance. We find however that astrocytes are more likely to form direct membrane apposition with ChIs than with SPNs, or PV-interneurons, an observation that we speculate could arise from the larger soma diameters of ChIs. Furthermore, our observation of slightly closer nearest astrocytes to the largest ChIs (Supplementary Fig. S5) suggests there may be a cell-cell anatomical association slightly closer than chance in NAcc.

Astrocytes are relatively well studied for close associations between their processes and synaptic terminals[56], but have received only limited attention for their proximity to neuronal somata[57], dendrites[58], or axons[59]. A soma-to-soma satellite configuration has been found between astrocytes and GABAergic interneurons in hippocampus CA1[53] but soma-soma localization has more typically been observed between neurons and oligodendrocytes[60–65] or microglia[66,67] for roles in local ion homeostasis[68]. A satellite configuration for astrocytes with neurons presents a distinct opportunity for astrocyte-neuron communication, and we show here that this arrangement for astrocytes and ChIs can be accompanied by powerful rapid regulation of ChI excitability.

### Astrocytes rapidly regulate ChI firing activity via regulation of extracellular Ca²⁺

We found that subsecond optogenetic depolarization of astrocytes in NAcc by a few millivolts resulted in rapid and reversible bidirectional changes in ChI excitability. Increases in spontaneous firing rates observed in ~30% of ChIs occurred with a short latency of milliseconds while decreases in firing rates observed in 60% of ChIs had longer onset latencies of several to tens of milliseconds. The rapid excitation responses are TTX-resistant revealing a direct mechanism independent of other neural circuits, whereas the slower inhibition responses are TTX-sensitive, supporting an indirect polysynaptic effect. We noted that membrane properties and single action potential characteristics were subtly different between the ChIs that showed excitation or inhibition. These intrinsic differences might be in keeping with discrete subpopulations of ChIs, or, might be generated by localized relationships to astrocytes. Different genetic subtypes of ChIs have yet to be identified definitively[69], but subpopulations of ChIs have been characterized by electrophysiological and morphological differences[47]. Whether satellite astrocytes are preferentially associated with one of these subpopulations remains to be determined. Diversity in astrocytes[18,70] might also contribute.

We established that the mechanism through which striatal astrocytes in NAcc excited ChIs did not require gap junctions, ionotropic glutamate or GABAergic transmission, or ATP/adenosine receptors, but involved rapid buffering of local $[Ca^{2+}]_o$. Modest changes in $[Ca^{2+}]_o$ (~ 0.1 mM) were detected by an extracellular microelectrode sensor placed locally to astrocytes, but will underrepresent any changes in $[Ca^{2+}]_o$ that occur closer to source mechanisms and within the intercellular clefts such as soma-to-soma appositions. The large surface area of soma-to-soma appositions could facilitate the impact of fluctuations in $[Ca^{2+}]_o$ driven by astrocytes on adjacent ChIs. ChIs express a plurality of Ca²⁺-sensitive currents that shape activity that are candidate mediators through which diminished $[Ca^{2+}]_o$ might promote depolarization and firing rates, including SK[50], BK[52], HCN[50] and potentially also NALCN currents. Preliminary analyses indicate that at least SK-operated mAHPs are reduced following astrocyte activation. We did not explore the mechanisms operating in DLS.

Roles for astrocytes in the regulation of extracellular [Ca²⁺] have only recently begun to be appreciated, to date in cortical layer V pyramidal neurons[30], and brainstem trigeminal nucleus[29] where astrocyte regulation of $[Ca^{2+}]_o$ regulates firing patterns. How can astrocytes rapidly and reversibly alter $[Ca^{2+}]_o$? There are many ready possibilities via Ca²⁺ channels expressed by astrocytes: Depolarization might lead

to $Ca^{2+}$ entry via astrocytic voltage-gated calcium channels ($Ca_v$)[18,71,72] or the astrocytic $Na^+$-$Ca^{2+}$ exchanger (NCX)[73] or $K^+$ currents might lead to $Ca^{2+}$ entry via $K^+$-operated NCKX[18]. Given that astrocytes generate intracellular $Ca^{2+}$ oscillations throughout their soma and extensive distal processes, and that these are at least partly supported by transmembrane flux of $Ca^{2+}$ [71], consequential changes to extracellular $[Ca^{2+}]$ seem likely and physiological. However, the modest $Ca^{2+}$ conductance of ChR2 might also contribute[74] and this cannot easily be excluded. Alternatively, there might be a role for astrocytic $Ca^{2+}$-binding protein S100β in buffering $[Ca^{2+}]_o$: activity-dependent release of S100β from astrocytes has recently been suggested to mediate changes to $[Ca^{2+}]_o$ that modulate trigeminal and cortical neurons[29,30], and since astrocytes in striatum are particularly enriched in S100β[18], it might play a role here, as yet untested. The possibility also remains that astrocyte depolarization might modulate ChIs via additional mechanisms, such as alterations in $[K^+]$[48], $[Zn^+]$ or nitric oxide signaling. In any event, the central involvement of $[Ca^{2+}]_o$ in the regulation of ChI excitability by astrocytes, together with the close apposition of these two cell types, could indicate that astrocytes regulate ChIs via a form of ephaptic coupling, an intercellular communication between adjacent excitable cells mediated by extracellular ion concentration changes that occur in a shared intercellular cleft[75]. The relatively large surface area of soma-soma appositions between astrocytes and ChIs might afford a privileged means through which small endogenous and local fluctuations in astrocytic transmembrane ion flux impact on extracellular $Ca^{2+}$ concentrations across a sufficiently large perimeter or location on the ChI plasma membrane to impact on somatic action potential generation in ChIs.

## Astrocytes change downstream networks and modify DA transmission

We observed that astrocyte depolarization had other effects on striatal networks besides excitation of ChIs: it inhibited other ChIs, and activated SPNs, with longer latencies than for excitation of ChIs suggesting mechanisms downstream of this early excitation. Both the inhibition of ChIs and activation of SPNs were TTX-sensitive and GABA-mediated, therefore involving additional circuits that release GABA. The inhibition of ChIs did not involve nAChRs, which can mediate polysynaptic ChI-ChI inhibition in dorsal striatum[45,46]. But a hypothesis remains that the ChIs excited at short latency by astrocytes could be a potential source/driver of GABA-dependent mechanisms: ChIs can co-release GABA[47], and can regulate SPNs via a GABA-receptor dependent mechanism involving nAChR-regulation of GABA interneuron activity[76]. Note that while GABA$_A$-receptors can excite SPNs ex vivo owing to a membrane potential more hyperpolarized than the reversal potential for the GABA$_A$-receptor-mediated chloride current[77], they would be expected to be inhibitory in vivo where membrane potential is more depolarized.

Astrocyte depolarization had strong effects on DA release. This rapid modulation was independent from the ability of striatal astrocytic transporters for GABA and adenosine to shape the level of tonic inhibition of DA release by GABA and adenosine A1 receptors[6,8]. Rather, it was mediated via nAChRs, which are expressed on DA axons[78] and powerfully gate striatal DA release[9,31], revealing a modulation downstream of ChI activation. We found that astrocyte depolarization reduced DA release evoked subsequently by single electrical pulses which in turn, supported activity-dependence of DA release for pulse trains, effects consistent with decreased nAChR activity[31,79]. Occasionally we noted that astrocyte activation directly activated DA release during astrocyte depolarization, particularly in NAcc, indicative of an increase in nAChR activation[9]. These seemingly paradoxical outcomes, of increased and then decreased nAChR activation are readily reconciled: they are consistent with an initial activation of nAChRs by ACh, which in NAcc reached threshold during astrocyte depolarization to locally drive DA release from axons, followed by the

well-known ability of nAChRs to desensitize after activation[80]. In vivo, the outcome of astrocyte depolarization on DA release could range from activation to inhibition depending on activity in the astrocyte and ChI networks. These findings reveal that the portfolio of actors through which astrocytes shape striatal DA, extends beyond tonic levels of GABA[6] and adenosine[8], to ACh dynamics.

## Diverse astrocyte-neuron communication mechanisms in health and disease

We reveal here that rapid astrocyte-neuron communication over a few milliseconds impacts broadly on striatal physiology and DA transmission, through rapid modulation of ChI activity by ephaptic-like coupling involving changes to $[Ca^{2+}]_o$. This timescale is at the most rapid end of intercellular signaling documented for astrocytes, which now spans from fastest timescale for ion flux, as seen here, to the slower timescales involving intracellular $Ca^{2+}$ signaling over seconds, through to molecular and structural plasticities over minutes. This diversity in roles of astrocytes could potentially reflect circuit-specialized functions for astrocytes[81]. Modest changes to astrocyte membrane polarization, which we show can be regulated over a small mV range by at least one striatal neurotransmitter receptor, might fluctuate moment-by-moment to shape ChI excitability according to striatal network activity.

The close anatomical relationship seen in striatum between astrocytes and ChIs, and the rapid timescale of responses of ChIs to extracellular $Ca^{2+}$ fluxes gated by astrocytes, create a rapid glia-neuronal signaling cascade that modulates ACh and downstream DA transmission. ChIs and DA are implicated in a broad range of neurological and psychiatric disorders such as Parkinson's disease, Huntington's disease, Tourette's disorder, schizophrenia, addiction, and depression, and these dynamic astrocyte-neuron interactions might provide new strategies to modify striatal circuit function in disease.

## Methods

All details of key lab materials used and generated in this study are listed in a Key Resources Table on Zenodo (https://doi.org/10.5281/zenodo.10553375). Protocols associated with this work can be found on protocols.io (https://doi.org/10.17504/protocols.io.dm6gp3k25vzp/v1).

### Mice

All procedures were performed in accordance with the Animals in Scientific Procedures Act 1986 (UK) (Amended 2012) with ethical approval from an Animal Welfare and Ethical Review Body at the University of Oxford, under the authority of Licenses (P9371BF54 and PP8860348) from the UK Home Office. Male and female mice were used at 8 to 16 weeks of age. Homozygous *ChAT-Cre* mice (Jax #006410) were crossed with C57BL/6 J (WT) mice to obtain heterozygous *ChAT-Cre* mice. Homozygous *ChAT-Cre* mice were crossed with homozygous *Ai32* mice (*Ai32 ROSA*$^{ChR2(H134R)-EYFP}$, Jax#024109) to obtain heterozygous *ChAT-Cre::Ai32* mice. Mice were group-housed and maintained on a 12 h light/dark cycle (lights on 07:00–19:00) with *ad libitum* access to food and water. All experiments were performed during the light phase of the cycle. Data from male and female mice were combined throughout as no difference in the effects of sex on astrocyte optogenetic activation on DA release (Supplementary Fig. S3) or ChIs (Supplementary Fig. S6) were observed.

### Intracranial Injections

Mice were anesthetized with 4–5% isoflurane ($O_2$ flow of -1 L/min) and subsequently maintained with -1.5–2% isoflurane during surgery. Mice were placed in a small animal stereotaxic frame (David Kopf 124 Instruments), and body temperature was maintained at -37 °C. After exposing the skull under aseptic techniques, a small burr hole was drilled and AAV solutions were injected at an infusion rate of 200 nL/min with a 32-gauge Hamilton syringe (Hamilton Company) using a

microsyringe pump (World Precision Instruments) and withdrawn 5 min after the end of injection.

To virally express the fluorescent proteins, receptors or channels, 500 nL per hemisphere of virus was injected unilaterally or bilaterally into nucleus accumbens (AP + 1.3 mm, ML ± 1.2 mm from bregma, DV − 3.75 mm from exposed dura mater) or dorsal striatum (AP + 0.8 mm, ML ± 1.75 mm from bregma, DV − 2.40 mm from exposed dura mater) of 8–12-week-old C57BL/6 J wildtype mice or *ChAT-Cre* mice. Mice were used for experiments 3–5 weeks post-intracranial injection. The following constructs were used (all obtained from ETH Zurich Viral Vector Facility): AAV5/hgfaABC$_1$D-ChR2(H134R)_EYFP ($7.8 \times 10^{12}$ genome copies per ml; v166-5); AAV5/hSyn1-DIO-mCherry ($1.3 \times 10^{13}$ genome copies per ml; v116-5); AAV5/hgfaABC$_1$D-eGFP ($7.3 \times 10^{12}$ genome copies per ml; v95-5); AAV5/hgfaABC$_1$D-mCherry ($6.1 \times 10^{12}$ genome copies per ml; v222-5); AAV5/hSyn1-DIO-hM4D(Gi)-mCherry ($5.6 \times 10^{12}$ genome copies per ml; v84-5).

## Fast-scan cyclic voltammetry

Acute brain slices for voltammetry were obtained from 8–16-week-old mice. Mice were culled by cervical dislocation within ~2 hrs after start of light ON period of light cycle and brains were dissected out and submerged in ice-cold cutting solution containing (in mM): 194 sucrose, 30 NaCl, 4.5 KCl, 1 MgCl$_2$, 26 NaHCO$_3$, 1.2 NaH$_2$PO$_4$, and 10 D-glucose. Coronal slices 300 μm-thick containing striatum were prepared from dissected brain tissue using a vibratome (VT1200S, Leica Microsystems) and transferred to a holding chamber containing artificial cerebrospinal fluid (aCSF) containing (in mM): 130 NaCl, 2.5 KCl, 26 NaHCO$_3$, 2.5 CaCl$_2$, 2 MgCl$_2$, 1.25 NaH$_2$PO$_4$ and 10 glucose. Sections were incubated at 34 °C for 15 min before they were stored at room temperature (20–22 °C) until recordings were performed. All recordings were obtained within ~8 h of slicing. All solutions were saturated with 95% O$_2$/5% CO$_2$. Before recording, individual slices were hemisected and transferred to a recording chamber and superfused at ~2.5–3.0 mL/min with aCSF at ~32–33 °C.

Individual slices were hemisected and transferred to a recording chamber and perfused with aCSF at ~32–33 °C at a rate of ~2.5–3.0 ml/min. A carbon fiber microelectrode (CFM; diameter ~7–10 μm; tip length 70–120 μm), fabricated in-house, was inserted 120 μm into the tissue and slices were left to equilibrate and the CFM to charge for ~45 min prior to recording. All experiments were carried out either in the dorsolateral quarter of the striatum (DLS) or nucleus accumbens core (NAcc), one site per slice. Evoked extracellular DA concentration ([DA]$_o$) was measured using FCV at CFMs as described previously[6]. In brief, a triangular voltage waveform was scanned across the microelectrode (−700 to +1300 mV and back versus Ag/AgCl reference, scan rate 800 V/s) using a Millar Voltammeter (Julian Millar, Barts and the London School of Medicine and Dentistry), with a sweep frequency of 8 Hz. Electrical stimuli were delivered to the striatal slices at 2.5 min intervals, which allow stable release to be sustained at ~90–95% over the time course of control experiments. Evoked currents were confirmed as DA by comparison of the voltammogram with that produced during calibration with applied DA in aCSF (oxidation peak at +500–600 mV and reduction peak at -200 mV). Currents at the oxidation peak potential were measured from the baseline of each voltammogram and plotted against time to provide profiles of [DA]$_o$ versus time. CFMs were calibrated *post hoc* in 2 μM DA in each experimental solution. Calibration solutions were made immediately before use from stock solution of 2.5 mM DA in 0.1 M HClO$_4$ stored at 4 °C. CFM sensitivity to DA was between 10 and 40 nA/μM.

For electrical stimulation, a local bipolar concentric Pt/Ir electrode (25 μm diameter; FHC Inc.) was placed ~100 μm from the CFMs and stimulus pulses (200 μs duration) were given at 0.6 mA (perimaximal in drug-free conditions). We applied either a single pulse or 5 pulses at 50 Hz.

For light stimulation, astrocytes were activated by TTL-driven (Multi Channel Stimulus II, Multi Channel Systems) brief pulses (500 ms) of blue light (470 nm; ~4–5 mW/mm; OptoLED, Cairn Research) which illuminated the full field of view (~500 μm diameter, 40x immersion objective), preceding the electrical stimulation by 400 ms. Unless otherwise specified, 3 trials of combined light stimulation/electrical stimulation with an intertrial interval of 2.5 min were interspersed with 3 trials of only electrical stimulation.

## Patch clamp electrophysiology

Acute brain slices for patch clamp electrophysiology were obtained from 8–16-week-old mice. Anesthesia was induced within ~2 hrs after start of light ON period of light cycle using intraperitoneal sodium pentobarbital (100 mg/kg) and mice were decapitated in ice-cold, NMDG-based cutting solution containing (in mM): 93 N-methyl-d-glucamine (NMDG), 93 HCl, 30 NaHCO$_3$, 25 D-glucose, 20 HEPES, 5 Na-ascorbate, 2 thiourea, 7 MgCl$_2$, 3 Na-pyruvate, 2.5 KCl, 1.25 NaH$_2$PO$_4$ and 0.5 CaCl$_2$ (310 mOsm, pH 7.4) oxygenated with 95% O$_2$/5% CO$_2$ before decapitation. For a subset of experiments (Supplementary Fig. S2k–o), slices were instead cut in artificial sucrose-based cutting solution containing (in mM): 86 NaCl; 2.5 KCl; 1.2 Na$_2$HPO$_4$; 25 Na$_2$HCO$_3$; 25 glucose; 75 sucrose; 0.5 CaCl$_2$ and 7 MgCl$_2$. After decapitation, the brain was quickly dissected on ice. Coronal 300 μm slices containing the striatum (bregma: +1.50 till +0.50 mm) were cut with a vibratome (Leica VT1200S) and incubated in cutting solution at 34 °C for 15 min., followed by oxygenated (95% O2/5% CO2) aCSF at 34 °C for 15 min. aCSF contained (in mM) 127 NaCl, 25 NaHCO$_3$, 25 D-glucose, 2.5 KCl, 1.25 NaH$_2$PO$_4$, 1.5 MgSO$_4$ and 1.6 CaCl$_2$. Slices were allowed to recover at room temperature for at least 1 hr before recordings.

Whole-cell recordings were made using borosilicate glass pipettes (3–6 MΩ resistance for neurons, 6–9 MΩ resistance for astrocytes) with intracellular solution containing (in mM) 120 K-gluconate, 10 KCl, 10 HEPES, 10 K-phosphocreatine, 4 ATP-Mg, and 0.4 GTP (pH was adjusted to 7.4 using KOH, and osmolarity measured ~290 mOsm). In a subset of experiments, 5 mg/ml biocytin was added for intracellular biocytin filling. For a subset of experiments (Supplementary Fig. S2k–o), patch pipettes (4–6 MΩ tip resistance) were filled with an intracellular solution containing (in mM): 130 K-Gluconate; 0.6 EGTA; 2 MgCl$_2$; 0.2 CaCl$_2$; 10 HEPES; 2 Mg-ATP; 0.3 Na$_3$-GTP, pH 7.3 with KOH (295–300 mOsm). Recordings were performed in aCSF at near-physiological temperatures (32–33 °C) using a Multiclamp 700B amplifiers and Digidata 1440 A acquisition board digitized at 20 kHz sampling rate. Data were acquired using Clampex 10.0.

Series resistance was typically <20 MΩ and fully compensated for bridge balance and capacitance; recordings in which the series resistance exceeded 20 MΩ were not included in the averages. No correction was made for liquid junction potential. Data analysis was performed offline using custom written Python software. For biocytin labeling, individual cells were recorded and filled for at least ~30 min.

Basic physiological characteristics were determined from voltage responses to square-wave current pulses of 500 ms duration, ranging from −100 pA to +400 pA, and delivered in 25 pA intervals for pSPNs, 750 ms in duration ranging from -200 pA to +300 pA in 25 pA steps for ChIs, and 500 ms in duration ranging from -200 pA to +1800 pA in 50 pA steps for astrocytes. Input resistance was determined by the slope of the linear regression through the voltage-current curve. Sag was determined as the ratio voltage difference between the lowest voltage response and the steady-state response at the last 100 ms to a square-wave current pulses of 750 ms duration from 0 pA to −200 pA. Spontaneous firing rate was calculated from the number of action potentials crossing 0 mV in the absence of any (positive or negative) input current. Single action potential (AP) characteristics were obtained from the first elicited action potential. AP threshold was defined as the inflection point at the foot of the regenerative upstroke,

where the first derivative exceeded 10 mV/ms. AP amplitude was defined as the voltage difference between the threshold and peak voltage. AP half-width was measured at half of the peak amplitude. The after-hyperpolarizing potential (AHP) amplitude was measured as the peak hyperpolarizing deflection from AP threshold following AP initiation. Spike frequency adaptation was determined from the first ISI and last ISI in response to a +100 pA square-wave current pulse of 750 ms duration. Electrophysiological characteristics are provided in Supplementary Tables S1-S2.

In experiments with light stimulation, astrocytes or ChIs were activated by TTL-driven (Multi Channel Stimulus II, Multi Channel Systems) brief pulses (2–500 ms) of blue light (470; ~4–5 mW/mm; OptoLED, Cairn Research) which illuminated the full field of view (~500 μm diameter, 40x immersion objective). Unless otherwise specified, we performed 20 trials of light activation with an intertrial interval of 10 s. Given minimal change over consecutive trials in response direction and magnitude (Supplementary Fig. S6c,d), responses were averaged into a single response per neuron.

To classify directional light responses of ChIs, we cross-referenced z-scored response magnitude of individual cells averaged over 20 trials to their baseline responses. Thresholds for classification were set at z-score +0.5 for excitatory responses, and z-score -0.5 for inhibitory responses. These values corresponded to minimal and maximal values observed at pre-light baselines (~1000 ms before light onset; Fig. 3) as well as minimal and maximal responses observed in light control experiments in absence of ChR2 (during 500 ms blue light; Supplementary Fig. S6a, b). ChIs exceeding -0.5 z-score negatively or +0.5 z-score positively were classified as 'inhibition' or 'excitation', respectively. Similarly, a z-score cut-off of -0.5 was used to classify post-light off-set effects as 'inhibition'.

Pharmacological puff applications were performed using borosilicate glass pipettes of similar size (~3–6 MΩ resistance), and located ~20 μm distance from as well as directly facing the recorded soma. Puff applications of 500–1000 ms in duration were administered through a custom pressure system. For glycine application experiments, glycine (10 mM backfill) was puffed locally at the surface of the slice near the recorded cell through a micropipette (4–6 MΩ tip resistance; 1 s puff duration) with a pneumatic drug ejection system [PicoPump PV 820 (WPI, Germany)] in presence or absence of strychnine (10 μM). Unless otherwise specified, we performed 5 applications per neuron with an inter-trial interval of 10 s. Voltage responses were averaged into a single response per neuron.

## Ca²⁺-selective microelectrodes

$Ca^{2+}$-selective microelectrodes were made as previously described[29,30]. Borosilicate glass electrode with tip opening of ~3–5 μm were pre-treated with dimethylchlorosilane and dried at 120 °C for 2 h. The tip of the pipette was backfilled with ~1–2 μl calcium ionophore I cocktail A (Sigma-Aldrich, St. Louis, Missouri, USA). The rest of the electrode was filled with ~15 μl $CaCl_2$ (2 M). Before and after recordings, the electrode was calibrated at experimental temperatures (~33 °C) with aSCF solutions containing increasing concentrations of calcium (0, 0.1, 0.2, 0.4, 0.8, 1.2, 1.6, and 2 mM), the potentials for each concentration recorded, and a calibration curve obtained fitted with a logarithmic function. All other comparison experiments ($Zn^{2+}$, $Mg^{2+}$) were performed in aCSF with 1.6 mM $Ca^{2+}$. Only electrodes exhibiting a potential jump exceeding 20 mV between concentrations 0.2 and 2 mM ('sensitivity') were used for experiments (mean: 37.8 ± 5.6 mV, $n = 8$ electrodes).

## GRAB$_{ACh}$ imaging

An Olympus BX51WI microscope equipped with a 470 nm OptoLED light system (Cairn Research), Iris 9 Scientific CMOS camera (Teledyne Photometrics), 525/50 nm emission filter (Cairn Research), and x10/0.3 NA water-immersion objective (Olympus) was used for wide-field fluorescence imaging of GRAB$_{ACh}$ in striatal slices. Image acquisition

was controlled using Micro-Manager 1.4. Electrical stimulations, LED light, and image acquisition were synchronised using TTL-driven stimuli via Multi Channel Stimulus II (Multi Channel Systems). Image files were analyzed with Matlab R2017b and Fiji 1.5.

For experiments measuring extracellular ACh levels in response to electrical stimulation, images were acquired at 10 Hz (100 ms exposure duration) during continuous blue light (470 nm; ~1 mW/mm; OptoLED, Cairn Research) for a 10 s recording window. We extracted fluorescence intensity from the region of interest (ROI; 100 × 100 μm) and derived a background subtracted fluorescence intensity ($F_t$) by subtracting background fluorescence intensity from an equal-sized ROI where there was no GRAB$_{ACh}$ expression (i.e. cortex). Data are expressed as a change in fluorescence ($\Delta F/F_0$) and were derived by calculating $[(F_t\text{-}F_0)/F_0]$, where $F_0$ is the average background-subtracted fluorescence intensity ($F_t$) of the second preceding electrical stimulation. Background-subtracted fluorescence intensity ($F_t$) was extracted from an ROI (150 × 150 μm) ~50 μm from the stimulating electrode and $\Delta F/F_0$ was derived by calculating $[(F_t\text{-}F_0)/F_0]$, where $F_0$ is the average fluorescence intensity over the 1 s window (10 images) prior to the onset of light stimulation.

## SR101 labeling

For SR101 labeling of wild-type astrocytes in non-transgenic brain slices, acute brain slices were prepared as described above. Following recovery, acute slices were incubated in 1 μM SR101 (CAS 60311-02-6, Tocris Bioscience) dissolved in aSCF oxygenated with 95% $O_2$/5% $CO_2$. After, slices were briefly rinsed in fresh aCSF and used for further experiments.

## Immunofluorescent labeling: Mouse Tissue

Deep anaesthesia was induced by intraperitoneal injection of pentobarbital, and mice were transcardially perfused with saline followed by ice-cold 4% paraformaldehyde (PFA). Brains were dissected and post-fixed in 4% PFA for 2 h at room temperature. Brains were transferred to phosphate buffer (PB 0.1 M, pH 7.3) and stored overnight at 4 °C. Fifty micrometer coronal sections were collected serially (rostral to caudal) using a vibratome (Leica VT1200S) and stored in 0.1 M PB.

Sections were pre-incubated with a blocking PBS buffer containing 0.5% Triton X-100 and 10% normal bovine serum (NBS; ab7479, Abcam) for 1 h at room temperature. Sections were incubated in a mixture of primary antibodies in PBS buffer containing 0.4% Triton X-100 and 2% NBS for 72 h at 4 °C. The following primary antibodies were used: Guinea pig anti-S100β (1:1000, 287004, Synaptic Systems; RRID:AB_2620025); Mouse anti-GFAP (1:1000, G6171, Sigma-Aldrich; RRID:AB_1840893); Goat anti-ChAT (1:100; AB114P, Millipore; RRID:AB_2313845); Rabbit anti-NeuN (1:300, ab104255, Abcam; RRID:AB_10716451); Mouse anti-PV (1000, P3088; Sigma-Aldrich; RRID:AB_477329); Chicken anti-GFP (1:1000, GFP-1010, Lot GFP3717982, Aves; RRID:AB_2307313). Sections were washed with PBS and incubated with corresponding Alexa-conjugated secondary antibodies (1:300, Invitrogen) and cyanine dyes (1:300, Sanbio, Uden, The Netherlands) in PBS buffer containing 0.4% Triton X-100, 2% NBS for 5 h at room temperature. For anti-PV immunofluorescence, 0.3% Triton X-100 was used instead of 0.4%, and without serum during primary or secondary antibody incubation. Sections were counterstained with 1:10000 DAPI in 0.1 M PB (MBD0015, Sigma-Aldrich) for 15 min, washed with PB 0.1 M, mounted on glass slides, cover slipped with Vectashield H1900 fluorescent mounting medium (Vector Labs, Peterborough, UK), sealed and imaged.

Recovery of biocytin-labeled cells following electrophysiological recordings was performed as reported before[82–84], with minor alterations. Specifically, 300 μm slices were incubated overnight at 4 °C in fresh 4% PFA. Slices were extensively rinsed at room temperature in PBS and stained in PBS buffer containing 0.4% Triton X-100, 2% normal bovine serum (NBS; ab7479; Abcam) and streptavidin-conjugated Cy3

(1:300; Invitrogen; for GFP-labeled cells) overnight at 4 °C. Slices were washed with PBS and PB 0.1 M and mounted on slides, cover slipped with Vectashield H1900 fluorescent mounting medium (Vector Labs, Peterborough, UK), sealed, and imaged (see Confocal Microscopy and Analysis).

## Immunofluorescent labeling: human tissue

Post-mortem human brain tissues from three healthy subjects (Case 1, age 82 years, male; Case 2, 78 years male; Case 3, age 70 years male; Supplementary Table 3) were provided by the Oxford Brain Bank (OBB). The OBB is embedded within the Oxford University Hospitals NHS Foundation Trust and University of Oxford as part of the NIHR Oxford Biomedical Research Center, and is part of the Brains for Dementia Research (BDR) network. It complies with the requirements of the Human Tissue Act 2004 and the Codes of Practice set by the Human Tissue Authority (HTA license number 12217). Post-mortem human tissue samples were collected in accordance with approved protocols by the Ethics Committee of the University of Oxford (ref 15/SC/0639). All participants had given prior written informed consent for the brain donation.

Six-μm thick sections of formalin-fixed paraffin-embedded (FFPE) tissues containing the anterior basal ganglia at the level of nucleus accumbens were collected, as detailed previously[85]. Tissue sections were placed in an oven at 70 °C for 30 min, dewaxed in xylene (3 × 5 min) and rehydrated through decreasing concentration of industrial denatured alcohol (IDA) (100%, 100%, 90%, 70%; 5 min each) and subsequently in distilled water (5 min). Next, tissue sections underwent heat-induced epitope retrieval with autoclave (121 °C, 20 min) in 0.01 M sodium citrate buffer (pH 6.0) before rinsing in PBS (3 × 5 min) and incubation at 4 °C overnight with a mixture of the following primary antibodies: anti-ChAT antibodies (1:50, AB144P; Millipore, UK; RRID:AB_2313845); anti-GFAP antibodies (1:2000; Z0334; Agilent Dako, Santa Clara, United States; RRID:AB_10013382). Antibodies were diluted in 0.3% Triton X-100, 2% fetal bovine serum (FBS) and PBS.

On the second day, tissue sections were rinsed in PBS (3 × 5 min) and incubated with a mixture of Alexa Fluor 488-conjugated donkey anti-rabbit secondary antibody (1:200; A-11055; ThermoFisher Scientific, UK) and Alexa Fluor 594-conjugated donkey anti-goat secondary antibody (1:200; ThermoFisher Scientific, UK) for 1 h at room temperature. Then, sections were incubated with TrueBlack® Lipofuscin Autofluorescence Quencher (1:20 with 70% ethanol; Biotium, Fremont, CA, United States) for 30 s to block endogenous fluorescence signal before washing in PBS (3 × 2 min) and cover slipped and mounting with Vectashield antifade mounting medium with DAPI (H1900, Vector Laboratories, Peterborough, UK).

## Confocal microscopy and analysis: mouse tissue

Confocal imaging was performed using a Zeiss LSM 980 microscope (Carl Zeiss) equipped with Plan-Apochromat 10x/0.45 NA and 63x/1.4 NA (oil immersion) objectives. DAPI, Alexa488/GFP and A558/Cy3/mCherry/tdTomato were imaged using excitation wavelengths of 405, 488, and 561, respectively.

To quantify transduction efficiency and specificity of *ChAT-Cre* and GFAP-promoter driven constructs, we attained single plane, 1 μm optical thickness/optical plane, overview images (3140 × 3140 μm; 3790 × 3790 pixels), centered over the striatum at ×10 magnification with 1x digital zoom with 300 μm intervals across the full rostrocaudal axis of striatal viral expression. Off-line, images were analyzed using the multi-count tool in FIJI software. In brief, all individual GFP-labeled somata across the striatum with intact somata were checked to contain a DAPI+ nucleus, and DAPI + /GFP+ somata were manually quantified for soma colocalization with S100β, NeuN, ChAT, or PV immunofluorescence.

For quantification of astrocyte proximity, we attained 30 μm thick z-stack images (512 × 512 pixels) at 63x magnification with 1x digital zoom, 1 μm optical thickness/optical plane, and a step size of 2 μm, centered over ChAT + , NeuN+ or PV+ cells while blinded for S100β or DAPI signal, in either DLS (<250 μm distance from corpus callosum) or NAcc (<250 μm distance from anterior commissure) over the rostrocaudal extent of the striatum (bregma +1.70 mm till +0.74 mm). For ChAT+ and PV+ cells, all intact cells (e.g. avoiding cut cells at slice edge) within the region of interested were selected and imaged. For NeuN+ cells, we overlayed a 100 × 100 μm counting grid across the region of interested and selected and imaged the single NeuN+ cell situated closed to the center, resulting in roughly equal ChAT+ cells and NeuN+ cells per area. Off-line, images were converted to a maximum projection using FIJI software, and a ROI was drawn around the center cell in ChAT/NeuN/PV fluorescence channel using the polygon selection tool, whilst blind to S100β channel and thus potential astrocyte location. Next, in the S100β fluorescence channel, a ROI was drawn around the closest astrocyte soma, defined as the S100β signal (> 5 μm in diameter) featuring a DAPI+ nucleus residing closest to center of the image (and thus center cell). From the ROIs, we obtained soma surface area and *x/y* coordinates, which were used to calculate intersoma distance and create the location density plots (Fig. 2). Distance between neurons and astrocyte was calculated using the center point of each cell. Direct soma-to-soma contact was defined as a minimal direct apposition of more than 3 μm in length of putative cell membranes of fluorescent cells. No spatial corrections were made for tissue shrinkage. The location density plots were created from the ROI coordinates using custom written Python code using a Gaussian filter (sigma = 24).

## Intercellular distance modeling

For mathematical predictions of geometrically expected mean minimum distances between ChIs or SPNs and their nearest astrocyte, we used mouse striatal cell densities as follows: NeuN-positive neurons (approximated to SPNs) of 16 per 100,000 μm³ from ref. 39; from relative density of astrocytes:SPNs of 1:6[18,39], a density of 2.67 astrocytes per 100,000 μm³; from relative density of ChIs:SPNs of 1:50[37], a density of 1 ChI in 312,500 μm³. Therefore, relative densities in 312,500 μm³ of SPNs:astrocytes:ChIs of 50: ~ 8.3:1. We used a range of mathematical models to calculate mean minimum spacing based on either: (i) a uniform lattice of cells with a geometric solution based on ChIs or SPNs located within cubes of known volumes to predict distance to astrocytes within the cube, with a variation to account for sampling in a thin section; or (ii) uniformly random distributions of cells within their cube, with Monte-Carlo style solutions, with a variation to ignore positions within too small a cell boundary distance (for equations and code see Supplementary Methods).

## Confocal microscopy and analysis: human tissue

Immunofluorescent-stained slides were visualised and imaged using Zeiss 880 Airyscan inverted confocal microscope (Carl Zeiss). Images were acquired using 10x objectives (Plan-Apochromat/0.45 NA) and 63x objectives (Plan-Apochromat, oil immersion; DIC M27/1.4 NA) with laser excitation at 405 nm (solid state), 488 nm (Argon), and 561 nm (solid state). Image capture and processing were performed using the Zen Black and Zen Blue (Carl Zeiss, Germany) software.

## Drugs

Drugs were diluted to their required concentrations in aCSF immediately before use.

Apamin (100 nM) was obtained from Calbiochem. AP-V (50 μM), (+)-bicuculline (10 μM), 8-cyclopentyl-1,3-dimethylxanthine (CPT, 10 μM) and L-741626 (1 μM) were obtained from Abcam. BAPTA (5 mM), caffeine (20 μM), carbenoloxone (CBX, 100 μM), CGP 55845 hydrochloride (CGP, 4 μM), dihydro-β-erythroidine hydrobromide (DHβE, 1 μM), EGTA (various), NBQX (5 μM), oxotremorine (10 μM),

riluzole hydrochloride (100 μM), Strychnine (10 μM), tetrodotoxin (TTX, 1 μM) and ZD7288 (50 μM) were obtained from Tocris. Suramin (100 μM) was obtained from Millipore. Glycine (10 mM) was obtained from Sigma-Aldrich.

### Statistical analysis

Statistical analysis was performed using GraphPad Prism v7. Sample sizes were chosen based on our previous work with similar techniques. Blinding was performed for histological experiments. Data sets were analyzed using the Shapiro–Wilk test for normality. No outlier data were identified or removed. Fast-scan cyclic voltammetry, anatomy and patch-clamp electrophysiology experiments were designed using sample sizes informed by previously published studies[6,84,86]. Data sets with normal distributions were analyzed for significance using paired or unpaired Student's two-tailed $t$ test or analysis of variance (ANOVA) measures followed by Tukey's *post hoc* test. Data sets with non-normal distributions were analyzed using Mann–Whitney $U$ test or Kruskal–Wallis test with Dunn's adjustment for multiple comparisons. Correlations were examined using Pearson's $r$ test. Data are expressed as mean ± standard error throughout. $n$ indicate *number of slices*(*number of mice*) for FCV, GRAB$_{ACh}$ imaging and Ca$^{2+}$-selective electrodes, and *number of cells*(*number of mice*) for anatomy and whole-cell electrophysiology. Images are representative images from at least three independent replications. Exact $P$-values values are provided throughout the text, except when $P < 0.001$. The significance threshold was set at $P < 0.05$.

### Reporting summary

Further information on research design is available in the Nature Portfolio Reporting Summary linked to this article.

## Data availability

Available on Zenodo (https://doi.org/10.5281/zenodo.10553375). Source data are provided with this paper.

## Code availability

The code used and generated in this study is available from GitHub (https://github.com/craggASAP/astrocytes.git) and Zenodo (https://doi.org/10.5281/zenodo.13883128).

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

## Acknowledgements

The work was funded by grants from Aligning Science Across Parkinson's (ASAP-020370) through the Michael J. Fox Foundation for Parkinson's Research (MJFF) to S.J.C. and P.J.M., and from the Medical Research Council awarded to S.J.C. and B.M.R. (MR/V013599/1), and P.J.M. (MC-UU-00003/5), a Christ Church Oxford Research Center Grant to S.J.C., and a Fellowship to B.M.R. from St John's College Oxford. We acknowledge the Oxford Brain Bank, and support from Brains for Dementia Research (BDR) (Alzheimer's Society and Alzheimer's Research UK) and the NIHR Oxford Biomedical Research Center to L.P. The authors would like to thank Dorly Verdier and Arlette Kolta (University of Montreal, CA) for providing the protocol and helpful suggestions with regards to calcium-selective electrodes. For the purpose of open access, the author has applied a CC BY 4.0 public copyright licence to all Author Accepted Manuscripts arising from this submission.

## Author contributions

Conceptualization: S.J.C., B.M.R., and J.S.; Methodology: J.S., B.M.R., S.R., S.B., A.K.L.L., L.P., N.M.D., K.M.; Investigation: J.S., B.M.R., S.R., S.B., A.K.L.L., N.M.D., K.M.; Writing—Original Draft: S.J.C. and J.S.; Writing—Review & Editing: B.R., S.R., A.K.L.L., N.M.D., K.M., P.J.M., L.P.; Resources: L.P., P.J.M., S.J.C.; Supervision: S.J.C., J.S., P.J.M., and L.P. Funding Acquisition: S.J.C., B.M.R., P.J.M.

## Competing interests

The authors declare no competing interests.

### Inclusion and diversity

We support inclusive, diverse, and equitable conduct of research.
