## [Transparent Peer Review file · Nature Communications]

Rapid modulation of striatal cholinergic interneurons and dopamine release by satellite astrocytes

Corresponding Author: Professor Stephanie Cragg

Version 0:

Reviewer comments:

Reviewer #1

(Remarks to the Author)

I read the paper from the Cragg group with great interest. I find the paper is beautifully written, the experiments are clear and the figures are logical and in keeping with the results.

The authors went on a significant detective search to identify why brief depolarization of striatal astrocytes causes cholinergic interneuron firing. They made several impressive and important discoveries. First, they found that a population of astrocytes sits opposed to cholinergic interneurons (this is new biology and potentially relevant for many other brain areas). Second, they dissected the mechanism by which these juxtaposed cells communicate through clever experiments and detective work, finding that astrocytes regulate neurons through changes in extracellular calcium ion concentrations. This is a novel mechanism for astrocyte-neuron interactions and underscores the realization that astrocyte subsets have distinct functions in neural circuits. One wonders how often this mechanism has been at play in the past over, but has been missed in past studies. From this perspective, the main finding of this study is highly significant. I also found the precision in thinking and modeling to be reassuring; this will be much valued by the readers. Overall, the authors unmasked a novel mechanism linking astrocytes, cholinergic interneurons and dopamine release in a critical area of the brain.

In my view, this is an excellent paper. I have no criticisms and I found no weaknesses in the work, writing, or data. The paper could be published as is.

My only minor suggestion is to increase the size of the fonts in some of the figures and perhaps to remove the boxes around the graphs (e.g. Fig 3e,f,g etc.). However, I realize this is a personal preference that the authors can decide for themselves.

Many congrats to all the authors and the Cragg lab on another excellent study.

Reviewer #2

(Remarks to the Author)

This manuscript by Stedehouder et al. investigate how striatal astrocytes regulate cholinergic interneurons (ChI) and dopamine release. For that aim, they use optogenetic activation of astrocytes using ChR2, electrophysiology, voltametry and a variety of chemo- and optogenetic tools. They propose that astrocyte photoactivation decreases extracellular calcium and regulate the firing rate of ChI with consequences on dopamine release in both the NAcc and the DLS. They propose that this fast regulation is possible thanks to the close proximity of astrocytes and ChI that they called "satellite astrocytes" and can be found also in human. The idea that astrocytes can regulate the activity of striatal local interneurons having an impact on dopamine release is novel, timely and interesting. However, the manuscript has mayor technical limitations which make it difficult to translate their results to physiological conditions.

1. The idea that astrocytes are electrically active is controversial. Glutamate transporter currents have been broadly described and characterized, however, these currents are local and in response to neuronal activity. As the expression of voltage-gated ion channels in astrocytes has not been demonstrated so far, astrocytes are still mainly considered non-electrically excitable cells. In this frame, the use of ChR2 to manipulate astrocytes is not suitable. The opening of ChR2 induces Ca²⁺ entry in the cytosol together with other ions (possibly Na) inducing a depolarization and a change in pH.

Indeed, the authors show that photoactivation of ChR2 induces a depolarization in astrocytes, but, can this or a similar depolarization be observed in striatal astrocytes under physiological conditions?

2. How astrocytes can control extracellular Ca²⁺ is not clear. As, ChR2 can permeate Ca²⁺, it is possible that the effect they observe is due to massive entry of Ca²⁺ into astrocytes, depleting it from the vicinity of the ChI. If this is the case, it is improbable that a similar mechanism exists in physiology. Can they reproduce the results by depolarizing patched astrocytes?

3. Different brain regions were used for experiments showing the involvement of ChI in the modulation of dopamine release (DLS) and the mechanism of action (NAc). They show that, in the DLS, astrocyte activation regulate dopamine release by activating ChI. Is this also the case in the NAc? Perhaps in the NAcc astrocytes can directly regulate dopamine release by releasing Ach or any other gliotransmitter.

4. In the same line, the authors found 3 types of ChI responses to astrocyte photoactivation in the NAcc, are these populations also found in the DLS or is it specific for the NAcc? And if so, is it through the same mechanism found in the NAc (extracellular calcium clearance)?

5. Can extracellular Ca²⁺ changes also regulate dopamine release in both the NAcc and the DLS?

6. The authors propose that the increase of firing rate in ChI is a direct astrocyte-to-ChI communication, while the decrease in firing rate is mediated by an indirect mechanism probably involving another intermediate cell type. Is it possible that the excited ChI are more proximal to satellite astrocytes than inhibited ChI?

Reviewer #3

(Remarks to the Author)

The authors explored the role of rapid activation of astrocytes (using optogenetics) in modulating dopamine release. This experiment took them down the proverbial "rabbit hole" as they discovered diverse and often opposite effects triggered by rapid depolarization of astrocytes. Nevertheless, they bravely and systematically work through the mechanisms of these effects using a broad range of clever genetic, anatomical and bleeding-edge electrophysiological and electrochemical approaches. This is a admirable tour-de-force and extremely well-controlled study.

I'll try to describe the main set of findings presently, but will first cut to the chase and mention that the central and exceptional finding is that many of the astrocytes effects on striatal circuitry are mediated by their ability to rapidly, and directly excite cholinergic interneurons (ChIs) by reducing extracellular calcium concentration in the vicinity of the ChIs, because, as the author's anatomical data suggest, astrocytes form unique satellite-like close appositions to ChIs.

so here goes:

First, the authors show the optogenetic activation of astrocytes reduces dopamine release (caused by electrical stimulation) and probability of release, except in the N. Accumbens (NAcc) where they can also trigger dopamine release (independently of electrical stimulation). In the discussion they ascribe this difference to two mechanisms that Cragg has published previously in two seminal studies. Direct activation of nAChRs (mainly in NAcc) and desensitization of nAChRs which leads to a reduction in DA release.

They then show that these effects are not GABA or adenosine receptor dependent.

However, blocking nAChRs or chemogenetically silencing ChIs prevents this effect.

Then, they provide evidence for the uniquely close apposition of astrocytes to CINs, and that this configuration is found in almost half of CINs (by comparison astrocytes are closely apposed to only 1/12 of putative spiny projection neurons).

Second, they show the optogenetic activation of astrocytes changes ChI firing rates bi-directionally: it can cause either fast excitation (within 10ms) or slow inhibition (on the order of 70 ms) of ChIs (whether a ChI is excited or inhibited is a property of the ChI and doesn't change). The fast excitation is TTX insensitive, and also insensitive to blockers of fast GABAergic and glutamatergic transmission. The slow inhibition is TTX-sensitive and GABAR dependent, and it is concluded that it is disynaptic via a GABAergic interneuron.

Anecdotally, they claim the ChIs can cause GABAR-dependent, disynaptic excitation of pSPN.

Finally, focusing on the fast excitation, they show that the main effect of rapid activation of astrocytes is to lower extracellular calcium and show that lowering of extracellular calcium excites ChI, thus they conclude, as stated above, that astrocytes modulate striatal dopamine release by reducing intracellular calcium concentration around cholinergic interneurons (CINs).

And as an epilogue, the authors provide evidence of the satellite phenotype in humans, highlighting the similar architecture of CIN-astrocyte proximity in the NAcc and putamen from postmortem brains.

Following are my major concerns:

1. Calcium changes of CIN excitability: How does extracellular calcium make cells more excitable? While reduction in extracellular calcium should slightly hyperpolarize its very depolarized reversal potential, it is hard to believe that this is a large enough effect to account for the effect (but this option should be mentioned). There is a hint to an answer in the characterization of various physiological measures of ChI excitability that distinguish between ChIs that are excited by

astrocytes and those that are inhibited. It turns out the mAHP is one of the most sensitive measures of this difference. So it seems likely that the reduction in calcium concentration causes the SK current that drives the mAHP to be smaller. This should be tested directly, either by measuring the SK in voltage clamp in response to activation or measuring the mAHPs during this excitation (as in Fig. 3e, top panel). In fact, the EGTA experiment (Fig. 5g) demonstrates that the main effect of chelating calcium is to reduce the mAHPs leading to a burst, lending support to this hypothesis, so this should be quantified as well.

2. Astrocytic satellites: You report that 43% percent of CINs have a satellite astrocyte and only 8% of pSPN. However, in absolute numbers (because there are two orders of magnitude more SPNs than CINs) the pSPN astrocytic satellites should outnumber the former by an order of magnitude. This suggests to me that the satellite configuration is common on many cells (could it change dynamically?) and perhaps there is a preference to CINs because they are large cells that perhaps need more "glial support" than the smaller SPNs. In any event, I think the fairest comparison should be to count these satellites on one of the GABAergic interneurons (GIN), and convince the reader that ChIs have more satellites than a class of GINs that is as numerous as CINs.

3. Which GIN?: The authors do not identify which GIN mediates the disinaptic inhibition of CINs by astrocytes or the disinaptic excitation of pSPN. Just as they silenced CINs to prove that the astrocytic effect on DA release depends on them, the authors should try to identify the GINs that are doing this by selectively silencing their best candidate for each of these mechanisms. And once they identify the GIN then test if to see if astrocytes are closely apposed to them (see previous comment), because this would generalize the mechanism they discovered to GINs as well, i.e. that astrocytes can directly excite a GIN to excite pSPNs or inhibit a CIN. In general, you don't report what is the resting membrane potential of the pSPNs in these experiments. Also, you show a single example of CINs causing depolarizing PSPs in pSPNs but claim a significant effect in the figure caption (S9). You don't say if this is GABAR or glutamate receptor dependent. There is an entire literature about disinaptic inhibition of SPNs by ChIs via NPY-NGF GINs, and a papers by Sabatini and Kreitzer about excitation of SPNs by glutamatergic transmission. You need to clarify if the excitation you mention in pSPNs by optogenetic excitation of CINs is something new or something akin to what has been published, and you need to show the statistics.

4. Functional implications: This study shows that fast activation of astrocytes can do all sort of interesting (often opposite) things in acute slices? But is there any evidence that astrocytes membrane potential changes that rapidly in a physiological context. It seems to me that perhaps you are driving a system that is meant to do something else, which leads to all sorts of interesting network effects, that are not physiological. I would guess, that perhaps it is the other way around. Astrocytes can express nAChRs (do the astrocytes you are studying express them?) perhaps astrocytes are there to respond to the release of ACh from ChI axons and then function as a feedback onto ChIs. Or perhaps when too many calcium-activated potassium currents lead to a reduction in the excitability of CINs, the astrocytes sense it and kick in to reduce those potassium currents (see comment 1 above). Therefore, I think in order to try to make sense of these findings, the authors should test what synchronous optogenetic release of acetylcholine from ChIs does to astrocytes membrane potential and/or calcium signaling. More generally, the authors should propose how and in what context may astrocytes kick in to produce variegated responses they to. Also, you mention they form a syncytium (and show dye flowing between them) so if under physiological conditions they do exhibit rapid changes in voltage are these synchronized among them?

more minor concerns?

5. In figure 1s there seems to be a residual effect when silencing CINs. Could it be that there is a direct effect of astrocytes on nAChRs on DA fibers?

6. What did activation of astrocytes during chemogenetic silencing of ChIs do to the excitatory response in NAcc?

7. Does puffing high calcium reduce the excitability of CINs (it would be nice to complement the EGTA experiment with this)?

8. I think "occluded" on p. 6 l.9 should be "prevented". Or is this an occlusion experiment which I did not understand?

9. ChIs don't have a resting membrane potential, so this measurement is not meaningful because if the ChI has one it's an artifact of whole cell recording.

10. p. 15 "for any single ChI or SPN, their mean nearest neighbor astrocyte is at a similar soma-soma distance, " seems to contradict what you claim on p. 7 that ChIs are close. please clarify.

11. Why are changes in firing rate reframed as z-scores? what do you gain or what problem is it solving?

Reviewer #4

(Remarks to the Author)

The manuscript by Stedehouder and colleagues is a thorough study of the effect of astrocytes activation on striatal dopamine release. In a series of experiments the authors show that astrocytes exert modulatory control over striatal cholinergic interneurons, thus modulating the activity of both DA axons and spiny projection neurons. Overall, the manuscript is well written, the experimental design is solid, and the findings are both novel and significant. I have some comments/suggestions that are listed below.

It would be important to investigate the effect of astrocyte depolarization on striatal DA release during basic electrical stimulation of the slice. Can the authors show the effect of DREADD-mediated inactivation of astrocytes on DA release evoked by electrical stimulation?

Fig. 4. It would help if spontaneous AP frequency in excited and inhibited ChIs (for experiments shown on both a and l panels) is also shown since changes in voltage sag and AHP are expected to alter spontaneous neuronal firing. Same for the cell capacitance as a proxy of cell morphology. Additionally, if HCN channels and AHP components are important, the authors should measure Ih and AHP current in the voltage clamp mode, which gives a more accurate assessment of the real channel activity.

Page 12. "These data indicate that rapid astrocytic modulation of pSPNs is indirect via a neuronal source of GABA release, which might include GABA-releasing ChIs." The statement is speculative without further experiments, including a demonstration of the effect of GABAa (and potentially AChR) blockade on pSPN response to optogenetic stimulation of ChI (as on Suppl Fig 9r).

MINOR:

Fig. 4 Legend: "Excited ChIs compared to inhibited ChIs showed (d) a trend to lower input resistance compared to inhibited ChIs..." Should read 'a trend to higher input resistance'.

Suppl Fig S2. SR101 staining needs to be added to the Methods.

Suppl Fig S5a&h. Shouldn't the X-axes read "Distance to the nearest astrocyte"?

Suppl Fig S11i-l legend. "Z-scored firing rate responses in an example excitation ChI under 500-ms blue light activation" should read 'inhibition ChI'.

Fig 4 legend title needs some editing.

Page 12. "The TTX-sensitive depolarization of pSPNs by astrocytes (see Supplementary Fig. 6)..." This should refer to Supplementary Fig. 9.

Reviewer #5

(Remarks to the Author)

Version 1:

Reviewer comments:

Reviewer #1

(Remarks to the Author)

This is a very nice study. I don't have any additional comments and think the authors have done an excellent job of addressing all the Reviewer comments from all four Reviewers.

Reviewer #2

(Remarks to the Author)

The authors have made some changes in the manuscript to downtone the idea that astrocytes are electrically active and they have shown that neuronal ACh is released in the NAcc in response to astrocyte optoactivation. Moreover, they demonstrate that astrocyte fast depolarizations occur in response to glycine, which is a fantastic addition to the manuscript. Unfortunately, they have not satisfactorily addressed 3 of my previous concerns:

1. The fact that Ca²⁺ influx through ChR2 can be happening.

The authors state that "only a small fraction of ChR2 current is carried by Ca²⁺ influx, as we cited on original P17, and therefore we deem this an unlikely contribution". However, the paper cited (Lin et al Bioph J), as well as Nagel et al PNAS, show that ChR2 indeed permeates Ca²⁺. Thus, it is possible that the calcium clearance is due to calcium entry through ChR2. If this is the case, I still think that it is improbable that such a mechanism exist in physiology. To rule out this possibility, I still think that they should either try to reproduce the results by depolarizing patched astrocytes, show that cytosolic calcium is not increasing in astrocytes upon optostimulation (in the presence of thapsigargin to rule out calcium coming from internal stores), or try to block some of the channels/exchangers they mention in the discussion (Cav, NCX or NCKX18).

2. Key experiments are missing in the NAcc

Although similar in cellular composition, the DLS and the NAc are distinct brain regions with differences in their inputs and outputs, dopaminergic innervation and involvement in behavior. Thus, one should not assume that the results found in the DLS will also apply in the NAc, and vice versa. I understand that repeating all experiments in both brain regions is redundant and it is not necessary. However, key experiments should be done to show that the same mechanism applies in both brain regions.

Since most of the work has been performed in the NAcc, they should demonstrate the involvement of Ch1 in dopamine release regulation. The observation that neuronal ACh is released upon astrocyte optoactivation is a good addition to the manuscript but it is not a causal demonstration for Ch1 involvement in dopamine release.

If the authors are not willing to perform more experiments in the DLS, they should state in the discussion that a different mechanism may exist in this brain region.

3. Can extracellular Ca²⁺ changes also regulate dopamine release?

The authors have not provided evidence for the role of extracellular Ca²⁺ on dopamine release.

They have shown that astrocyte optoactivation can regulate dopamine release in both the NAcc and the DLS, in the DLS this requires Ch1 activation. However, the involvement of Ch1 in dopamine regulation in the NAcc has not been demonstrated in this manuscript.

On the other hand, they show that astrocyte optoactivation regulate the activity of Ch1 in the NAcc by decreasing extracellular Ca²⁺ levels. Therefore, it is not directly explored whether the regulation of dopamine release is due to a decrease in extracellular Ca²⁺ levels or through a totally different mechanism.

Reviewer #3

(Remarks to the Author)

As I pointed out in the first round of reviews the manuscript provides intriguing new observations regarding astrocytic control of cholinergic interneurons' control of dopamine release.

The reviewers suggested several reasonable requests for experiments in order to clarify either the mechanism, the physiological relevance or the interpretation of the findings. The authors have turned down many of the suggested experiments.

For a journal like Nature Communications the authors should (and the reviewers do) expect additional experiments and leave to a bare minimum the experiments that are to be refused.

I therefore think that the author need to conduct at the bare minimum the following two experiments that were suggested in the first round of reviews before I can consider the manuscript further.

I think reviewer 2's request to test whether depolarizing patched astrocytes can lead to a similar effect as that observed with ChR2 is a pertinent and easily achievable given that the authors are able to patch these cells. Particularly, since a strong depolarization of a single neuron by tens of millivolts should simultaneously depolarize many other astrocytes in the syncytium by a few millivolt so they should get the same result. If the experiments fails there is no harm, but if it works it will be a much stronger and physiologically relevant result if a localized cluster of astrocytes can generate the same effect

The statistical modeling and analysis of the authors regarding the privileged proximity of CINs to astrocytes is nice, but it's hard to judge its veracity. This is why I asked to conduct a similar analysis regarding the proximity of astrocytes to another class of GABAergic interneurons (GINs). Either way this experiment comes out would be valuable to the authors and the readers, and I don't see why the authors refuse to do it. If it turns out that astrocytes are as close to GINs that could explain the polysynaptic inhibition of CINs (if astrocytes can also directly excite GINs). It is really critical to find out whether this mechanism is unique to CINs or more general, so I think that addressing the possibility that GINs have a privileged status is critical.

In summary, because the phenomena are interesting but also very variegated, conducting more experiments is warranted to help clarify what is going on.

Reviewer #4

(Remarks to the Author)

The authors addressed all of my questions/concerns and I recommend the manuscript for publication.

Reviewer #5

(Remarks to the Author)

Version 2:

Reviewer comments:

Reviewer #2

(Remarks to the Author)

The authors have now performed experiments in the NAc providing a more complete story in this brain region, which may

similarly occur the DLS.

On the other hand, once again, the authors have overlooked my main criticism. The whole idea that astrocytes regulate neuronal firing by buffering calcium is challenged by the fact that ChR2 can permeate calcium. I can still argue that what they have observed is simply an artifact. In my opinion, one can speculate what is the mechanism of a phenomenon when multiple possibilities exist. However, if one of those possibilities is that the findings could be an artifact of the chosen method, it becomes essential to perform proper controls.

If the authors believe that calcium is entering through channels, blocking those channels with a cocktail of antagonists and assessing astrocyte calcium while activating them with ChR2 is such an easy experiment that I genuinely cannot understand why they refuse to do it. It even makes me think that they have tried and the experiment didn't work.

Notably, according to the response to Reviewer 3, the authors have attempted one of the experiments I suggested to address this issue: depolarizing astrocytes and monitoring firing activity in Chl. However, these experiments have not yet been successful.

The authors argue that this point is relevant for the field and not only for their manuscript. I fully agree, but, just because other people do it doesn't mean it is correct. In fact, other papers using ChR2 in astrocytes combine this technique with other complementary approaches or control experiments to demonstrate that the observed effects are not artifacts, unfortunately this is not always the case.

Reviewer #3

(Remarks to the Author)

I appreciate the additional experiments on PV interneurons, which strengthen the manuscript. I have no more concerns. Congratulations on an intriguing set of findings.

Reviewer #5

(Remarks to the Author)

Stedehouder et al NCOMMS-24-03524

REVIEWER COMMENTS and RESPONSES

Reviewer #1

I read the paper from the Cragg group with great interest. I find the paper is beautifully written, the experiments are clear and the figures are logical and in keeping with the results.

The authors went on a significant detective search to identify why brief depolarization of striatal astrocytes causes cholinergic interneuron firing. They made several impressive and important discoveries. First, they found that a population of astrocytes sits opposed to cholinergic interneurons (this is new biology and potentially relevant for many other brain areas). Second, they dissected the mechanism by which these juxtaposed cells communicate through clever experiments and detective work, finding that astrocytes regulate neurons through changes in extracellular calcium ion concentrations. This is a novel mechanism for astrocyte-neuron interactions and underscores the realization that astrocyte subsets have distinct functions in neural circuits. One wonders how often this mechanism has been at play in the past over, but has been missed in past studies. From this perspective, the main finding of this study is highly significant. I also found the precision in thinking and modeling to be reassuring; this will be much valued by the readers. Overall, the authors unmasked a novel mechanism linking astrocytes, cholinergic interneurons and dopamine release in a critical area of the brain.

In my view, this is an excellent paper. I have no criticisms and I found no weaknesses in the work, writing, or data. The paper could be published as is.

My only minor suggestion is to increase the size of the fonts in some of the figures and perhaps to remove the boxes around the graphs (e.g. Fig 3e,f,g etc.). However, I realize this is a personal preference that the authors can decide for themselves.

Many congrats to all the authors and the Cragg lab on another excellent study.

> We thank Reviewer 1 for their kind remarks and enthusiasm for the study.

Reviewer #2

This manuscript by Stedehouder et al. investigate how striatal astrocytes regulate cholinergic interneurons (ChI) and dopamine release. For that aim, they use optogenetic activation of astrocytes using ChR2, electrophysiology, voltammetry and a variety of chemo- and optogenetic tools. They propose that astrocyte photoactivation decreases extracellular calcium and regulate the firing rate of ChI with consequences on dopamine release in both the NAcc and the DLS. They propose that this fast regulation is possible thanks to the close proximity of astrocytes and ChI that they called "satellite astrocytes" and can be found also in human. The idea that astrocytes can regulate the activity of striatal local interneurons having an impact on dopamine release is novel, timely and interesting. However, the manuscript has mayor technical limitations which make it difficult to translate their results to physiological conditions.

1. The idea that astrocytes are electrically active is controversial. Glutamate transporter currents have been broadly described and characterized, however, these currents are local and in response to neuronal activity. As the expression of voltage-gated ion channels in astrocytes has not been demonstrated so far, astrocytes are still mainly considered non-electrically excitable cells. In this frame, the use of ChR2 to manipulate astrocytes is not suitable. The opening of ChR2 induces Ca²⁺ entry in the cytosol together with other ions (possibly Na) inducing a depolarization and a change in pH. Indeed, the authors show that photoactivation of ChR2 induces a depolarization in astrocytes, but, can this or a similar depolarization be observed in striatal astrocytes under physiological conditions?

> We thank Reviewer 2 for their comments. We are not suggesting that astrocytes are intrinsically electrically active. Rather, as we had stated (p3, para 2; p4 penultimate para), previous literature suggests a range of ion channels, transporters and exchangers can operate to polarize astrocyte membranes, some in response to neuronal inputs, leading to small membrane depolarizations of the magnitude observed here. We are not necessarily suggesting that the mechanisms we find in striatum need to be voltage-gated; the ion flux resulting from normal ion buffering could support local changes in membrane potential, as seen for glutamate transporter currents. Intriguingly however, striatal astrocytes do show a strong transcriptomic signature for many voltage-operated channels including for Ca²⁺, Na⁺, K⁺ (cacna/g, scn1a, kcna/j subunits - see Chai et al 2017 <https://doi.org/10.1016/j.neuron.2017.06.029>) raising the possibility that gating might be voltage-operated. It remains uncontroversial however that astrocytes have a hyperpolarised membrane potential and can vary their membrane potential locally in response to ion flux and current injection. We show that only small mV changes mediate the effects we observe. Consequently, we have **moderated the text for greater clarity**. On p3, we have: replaced “*electrically active*” with “*can vary their membrane potential*”; we have added a citation to the presence of voltage-gated channels; and we refer to how the presence of ion channels “*could influence local membrane potential*”.

Furthermore, regarding whether such a ChR2-induced depolarization can occur under physiological conditions, we cited on P4 several other instances where endogenous afferent inputs drive similar levels of depolarization. We are finding in parallel projects currently in progress that a range of different striatal modulators can depolarize striatal astrocytes. We have therefore **performed additional experiments** and **included additional new data** to show an example in which brief local application of a striatal neuromodulator (glycine acting at classical glycine receptors) drives large inward currents that result in depolarizations of a few mVs (**see Suppl Fig S2, new panels k-o**) indicating that striatal neuromodulators can transiently influence astrocyte membrane potential.

2. How astrocytes can control extracellular Ca²⁺ is not clear. As, ChR2 can permeate Ca²⁺, it is possible that the effect they observe is due to massive entry of Ca²⁺ into astrocytes, depleting it from the vicinity of the ChI. If this is the case, it is improbable that a similar mechanism exists in physiology. Can they reproduce the results by depolarizing patched astrocytes?

> We agree that it remains incompletely resolved how astrocytes control extracellular Ca²⁺ but there are precedents in the literature, that we cited in the original manuscript on P17. Only a small fraction of ChR2 current is carried by Ca²⁺ influx, as we cited on original P17, and therefore we deem this an unlikely contribution. We consider an intermediate and amplifying step more likely, such as for example Na⁺/Ca²⁺ exchanger (NCX). We cite previous findings on P17 that “*Depolarization might lead to Ca²⁺ entry via astrocytic voltage-gated calcium channels (Ca_v)^{18,71,72} or the astrocytic Na⁺-Ca²⁺ exchanger (NCX)⁷³ or K⁺-operated NCKX¹⁸*”. We have also **added an additional sentence** to this discussion point, on

P18, to offer an additional consideration that *“Furthermore, given that astrocytes generate intracellular Ca^{2+} oscillations throughout their soma and extensive distal processes, and that these are at least partly supported by transmembrane flux of Ca^{2+} ,⁷¹ physiological changes to $[Ca^{2+}]_o$ seem likely.”*

3. Different brain regions were used for experiments showing the involvement of ChI in the modulation of dopamine release (DLS) and the mechanism of action (NAc). They show that, in the DLS, astrocyte activation regulate dopamine release by activating ChI. Is this also the case in the NAcc? Perhaps in the NAcc astrocytes can directly regulate dopamine release by releasing Ach or any other gliotransmitter.

> Following initial experiments in both DLS and NAcc in Fig. 1, our physiological investigations were pursued exclusively in NAcc. To address the reviewer’s question about whether in NAcc it is astrocytes or neurons that are releasing ACh, we have **performed additional experiments** to directly investigate whether astrocytic activation in NAcc leads to a source of ACh release from neurons (ChIs), as we propose from data in DLS. We therefore combined optogenetic activation of astrocytes in NAcc with detection of ACh using the genetically encoded fluorescent sensor GRAB_{ACh}, and assessed whether ACh release was TTX-sensitive, indicating a source from neurons. Indeed, we confirm that detected ACh release was completely TTX-sensitive, indicating neuronal activation and subsequent release of ACh. This new data is now included as **new data panels Suppl. Fig. S3j-l**, along with **description of these new Results** on P6-7.

4. In the same line, the authors found 3 types of ChI responses to astrocyte photoactivation in the NAcc, are these populations also found in the DLS or is it specific for the NAcc? And if so, is it through the same mechanism found in the NAc (extracellular calcium clearance)?

> As indicated above, for the purpose of pursuing the underlying mechanisms we focussed on characterising only one region of striatum, the NAcc, and so do not have an entire duplicate data set in a second region, the DLS. That data set would be a bonus but would double the quantity of data which we consider beyond the scope of this study.

5. Can extracellular Ca^{2+} changes also regulate dopamine release in both the NAcc and the DLS?

> Again, in order to invest our efforts in elucidation of the underlying mechanisms, we focussed on the NAcc only, and did not explore changes to extracellular Ca^{2+} in DLS. We agree that a comparison of NAcc and DLS would certainly be an interesting future avenue.

6. The authors propose that the increase of firing rate in ChI is a direct astrocyte-to-ChI communication, while the decrease in firing rate is mediated by an indirect mechanism probably involving another intermediate cell type. Is it possible that the excited ChI are more proximal to satellite astrocytes than inhibited ChI?

> We agree this is certainly a tantalising possibility, but one we currently have no definitive answer for. This is something we hope to address in future work.

Reviewer #3

The authors explored the role of rapid activation of astrocytes (using optogenetics) in modulating dopamine release. This experiment took them down the proverbial “rabbit hole” as they discovered diverse and often opposite effects triggered by rapid depolarization of astrocytes. Nevertheless, they bravely and systematically work through the mechanisms of these effects using a broad range of clever genetic, anatomical and bleeding-edge electrophysiological and electrochemical approaches. This is an admirable tour-de-force and extremely well-controlled study.

I’ll try to describe the main set of findings presently, but will first cut to the chase and mention that the central and exceptional finding is that many of the astrocytes effects on striatal circuitry are mediated by their ability to rapidly, and directly excite cholinergic interneurons (ChIs) by reducing extracellular calcium concentration in the vicinity of the ChIs, because, as the author’s anatomical data suggest, astrocytes form unique satellite-like close appositions to ChIs.

so here goes:

First, the authors show the optogenetic activation of astrocytes reduces dopamine release (caused by electrical stimulation) and probability of release, except in the N. Accumbens (NAcc) where they can also trigger dopamine release (independently of electrical stimulation). In the discussion they ascribe this difference to two mechanisms that Cragg has published previously in two seminal studies. Direct activation of nAChRs (mainly in NAcc) and desensitization of nAChRs which leads to a reduction in DA release.

They then show that these effects are not GABA or adenosine receptor dependent.

However, blocking nAChRs or chemogenetically silencing ChIs prevents this effect.

Then, they provide evidence for the uniquely close apposition of astrocytes to CINs, and that this configuration is found in almost half of CINs (by comparison astrocytes are closely apposed to only 1/12 of putative spiny projection neurons).

Second, they show the optogenetic activation of astrocytes changes ChI firing rates bi-directionally: it can cause either fast excitation (within 10ms) or slow inhibition (on the order of 70 ms) of ChIs (whether a ChI is excited or inhibited is a property of the ChI and doesn’t change). The fast excitation is TTX insensitive, and also insensitive to blockers of fast GABAergic and glutamatergic transmission. The slow inhibition is TTX-sensitive and GABAR dependent, and it is concluded that it is disynaptic via a GABAergic interneuron.

Anecdotally, they claim the ChIs can cause GABAR-dependent, disynaptic excitation of pSPN.

Finally, focusing on the fast excitation, they show that the main effect of rapid activation of astrocytes is to lower extracellular calcium and show that lowering of extracellular calcium excites ChI, thus they conclude, as stated above, that astrocytes modulate striatal dopamine release by reducing intracellular calcium concentration around cholinergic interneurons (CINs).

And as an epilogue, the authors provide evidence of the satellite phenotype in humans, highlighting the similar architecture of CIN-astrocyte proximity in the NAcc and putamen from postmortem brains.

> We thank Reviewer 3 for their kind remarks, enthusiasm, and for the extensive interpretation, discussion and comments.

Following are my major concerns:

1. Calcium changes of CIN excitability: How does extracellular calcium make cells more excitable? While reduction in extracellular calcium should slightly hyperpolarize its very depolarized reversal potential, it is hard to believe that this is a large enough effect to account for the effect (but this option should be mentioned). There is a hint to an answer in the characterization of various physiological measures of ChI excitability that distinguish between ChIs that are excited by astrocytes and those that are inhibited. It turns out the mAHP is one of the most sensitive measures of this difference. So, it seems likely that the reduction in calcium concentration causes the SK current that drives the mAHP to be smaller. This should be tested directly, either by measuring the SK in voltage clamp in response to activation or measuring the mAHPs during this excitation (as in Fig. 3e, top panel). In fact, the EGTA experiment (Fig. 5g) demonstrates that the main effect of chelating calcium is to reduce the mAHPs leading to a burst, lending support to this hypothesis, so this should be quantified as well.

> There are many potential mechanisms through which extracellular Ca^{2+} could change ChI excitability, and we considered that these might operate plurally, making clean delineation a complex task. We state in the original manuscript Discussion that “ChIs express a range of Ca^{2+} -sensitive currents that shape activity that are candidate mediators through which diminished $[Ca^{2+}]_o$ would promote depolarization and firing rates, including BK^{70} , SK^{50} , HCN^{50} and potentially also NALCN currents.” We agree with the Reviewer that modulation of the SK channel is an important possibility among these. We were reluctant to draw too much focus to any single potential mechanism, but encouraged by the Reviewer, we have now quantified the mAHP directly before and during astrocyte-induced excitation as well as direct depolarization of ChIs by calcium chelation, and highlighted a potential role for SK channels. As the reviewer noted, the mAHP amplitude is smaller after both of these manipulations, consistent with a reduced SK current. We have therefore included an **additional paragraph** in the Results, on P15, included the analysis in an additional new **Supplementary Figure, S14**, and revised the Discussion on P18 to indicate that *“Preliminary analyses indicate that at least SK-operated mAHPs are reduced following astrocyte activation.”*

2. Astrocytic satellites: You report that 43% percent of CINs have a satellite astrocyte and only 8% of pSPN. However, in absolute numbers (because there are two orders of magnitude more SPNs than CINs) the pSPN astrocytic satellites should outnumber the former by an order of magnitude. This suggests to me that the satellite configuration is common on many cells (could it change dynamically?) and perhaps there is a preference to CINs because they are large cells that perhaps need more “glial support” than the smaller SPNs. In any event, I think the fairest comparison should be to count these satellites on one of the GABAergic interneurons (GIN), and convince the reader that ChIs have more satellites than a class of GINs that is as numerous as CINs.

> We agree that the size of ChIs might be a critical factor here, and our mathematical modelling of expected cell-cell distances suggests ChIs, as much larger cells, are likely to have more membrane apposition with astrocytes than smaller cells. Some of our empirical data (e.g. Suppl. Fig. S5a) also show an inverse relationship between proximity of cell centres and size of the accompanying CIN, suggesting there may an association closer than chance. We have now **revised the related text** to this effect in our Discussion on P16. This may relate to metabolic needs, a topic for a potential future study!

With respect to other neurons, we have only examined the presence of satellite astrocytes in relation to CINs, and for a comparison, pSPNs. The possibility remains that satellite astrocytes exist around other

interneurons as well, but was not a focus of our study as not critical for the significance of the main findings of our manuscript. We have replaced a single inadvertent use of the word “unique” to describe the interaction between astrocytes and ChI with the word “privileged”, as used throughout the rest of the MS.

3. Which GIN?: The authors do not identify which GIN mediates the disynaptic inhibition of CINs by astrocytes or the disynaptic excitation of pSPN. Just as they silenced CINs to prove that the astrocytic effect on DA release depends on them, the authors should try to identify the GINs that are doing this by selectively silencing their best candidate for each of these mechanisms. And once they identify the GIN then test if to see if astrocytes are closely apposed to them (see previous comment), because this would generalize the mechanism they discovered to GINs as well, ie. that astrocytes can directly excite a GIN to excite pSPNs or inhibit a CIN.

In general, you don't report what is the resting membrane potential of the pSPNs in these experiments. Also, you show a single example of CINs causing depolarizing PSPs in pSPNs but claim a significant effect in the figure caption (S9). You don't say if this is GABAR or glutamate receptor dependent. There is an entire literature about disynaptic inhibition of SPNs by ChIs via NPY-NGF GINs, and a papers by Sabatini and Kreitzer about excitation of SPNs by glutamatergic transmission. You need to clarify if the excitation you mention in pSPNs by optogenetic excitation of CINs is something new or something akin to what has been published, and you need to show the statistics.

> Thanks for useful critique of these various points.

We have not tried to identify the neuron type that might function as intermediary to inhibit a subset of ChIs. Our data suggests that it is mediated by GABA (blocked by GABA_A antagonists, Suppl. Fig. 10m), and involves Na⁺ channel activation (blocked by TTX, Suppl. Fig. 10j.). This could implicate a GIN, but not necessarily so. We cannot exclude that excited populations of ChIs are the intermediary neurons, as ChIs can corelease GABA. Our focus here rather was to identify the most rapid responses of ChIs, and to reveal that the most rapid and direct ‘upstream’ effects of excitation are mediated via an ionic mechanism. It was beyond the scope of this study and the principal findings we wanted to convey to also identify the circuits that mediate the many ‘downstream’ responses of all striatal neurons and interneurons. We have **revised the Discussion paragraph on P18-19** to more clearly set the scope and these potential hypotheses in context.

For pSPN recordings, we have now **added a new Suppl. Table S2** which contains electrophysiological characteristics of our pSPN population.

As to whether ChIs depolarize pSPNs via GABA or glutamate receptors (P12, and Suppl. Fig S9r-s):- We had explored in Fig S9r-s simply whether activation of ChIs in our hands might support a similar activation of pSPNs to that observed after astrocyte activation (Fig S9), which it did, and agree that the data would need more pharmacological characterisation to be able to draw out more specific conclusions. Those data in panels S9r-s added no additional information than already published in work from Faust/Koos/Higley/Sabatini etc. We therefore agree with the Reviewer that these supplementary data have limited value without further validation and have therefore **removed data panels S9r,s from Fig S9**, and from the text, and instead have cited Faust et al 2016 in the **significantly revised Discussion paragraph on P18-19** for showing that ChIs “*regulate SPNs via a GABA-receptor dependent mechanism involving nAChR-regulation of GABA interneuron activity*”⁷⁶.

4. Functional implications: This study shows that fast activation of astrocytes can do all sort of interesting (often opposite) things in acute slices? But is there any evidence that astrocytes membrane potential changes that rapidly in a physiological context. It seems to me that perhaps you are driving a system that is meant to do something else, which leads to all sorts of interesting network effects, that are not physiological. I would guess, that perhaps it is the other way around. Astrocytes can express nAChRs (do the astrocytes you are studying express them?) perhaps astrocytes are there to respond to the release of ACh from ChI axons and then function as a feedback onto ChIs. Or perhaps when too many calcium-activated potassium currents lead to a reduction in the excitability of CINs, the astrocytes sense it and kick in to reduce those potassium currents (see comment 1 above). Therefore, I think in order to try to make sense of these findings, the authors should test what synchronous optogenetic release of acetylcholine from ChIs does to astrocytes membrane potential and/or calcium signaling. More generally, the authors should propose how and in what context may astrocytes kick in to produce variegated responses they to. Also, you mention they form a syncytium (and show dye flowing between them) so if under physiological conditions they do exhibit rapid changes in voltage are these synchronized among them?

> We thank the Reviewer for posing an intriguing set of suggestions. In line with point 1 raised by Reviewer 2 about changes to astrocyte membrane potential, please see our response to Reviewer 2 which details the **new data** we have included to show indeed that a striatal neuromodulator receptor can modulate astrocyte membrane potential. Regarding the question of the reciprocal regulation of astrocytes by ACh in particular, this is an interesting possibility, but a different question and focus from the work presented here. At this point we are seeking to explore how astrocytes shape ChIs. There may yet turn out to be a reciprocal interaction, but that is an entirely separate study!

As suggested, we have also included **an additional new sentence about the context** in which we speculate astrocyte depolarization might operate, in the Discussion on P20. We speculate that *“Modest changes to astrocyte membrane polarization, which we show can be regulated over a small mV range by at least one striatal neurotransmitter receptor, might fluctuate moment-by-moment to shape ChI excitability according to striatal network activity.”*

More minor concerns

5. In figure 1s there seems to be a residual effect when silencing CINs. Could it be that that there is a direct effect of astrocytes on nAChRs on DA fibers?

> We agree that it appears in Fig. 1s that some residual effect has remained after chemogenetic (e.g. DREADD-mediated) silencing of ChIs. Although we have not tested this directly, our assumption is that chemogenetic inhibition of ChIs would limit but not necessarily prevent all firing activity in ChIs, especially during a combined electrical or astrocytic stimulus. It is highly likely that a small subset of ChIs will still show some breakthrough excitation after additional astrocyte optogenetic activation, making it difficult to draw conclusions about residual effects of astrocytes directly on DA fibres. We would also like to point out that there is a more complete occlusion of the effect after a muscarinic agonist under oxotremorine (10 μ M) in **Suppl. Fig. S3h**, arguing against a residual direct effect.

6. What did activation of astrocytes during chemogenetic silencing of ChIs do to the excitatory response in NAcc?

> This is an interesting question but something that we have not examined in the current study.

7. Does puffing high calcium reduce the excitability of CINs (it would be nice to complement the EGTA experiment with this)?

> Our specific hypotheses concerned a reduction in extracellular calcium, and so we specifically tested manipulations that would lower Ca^{2+} . We had no specific hypothesis concerning high Ca^{2+} and thus is not something we have examined.

8. I think “occluded” on p. 6 l.9 should be “prevented”. Or is this an occlusion experiment which I did not understand?

> We thank the Reviewer for this correction. We have now **changed** the wording to “prevented”.

9. ChIs don't have a resting membrane potential, so this measurement is not meaningful because if the ChI has one it's an artifact of whole cell recording.

> We are happy to remove the respective data from Main Fig. 4. We had quantified ‘membrane potential’ as the average membrane potential in between action potentials. We have now **removed this panel** from the **Main Fig. 4**.

10. p. 15 “for any single ChI or SPN, their mean nearest neighbor astrocyte is at a similar soma-soma distance, “ seems to contradict what you claim on p. 7 that ChIs are close. please clarify.

> We are happy to clarify! We have **modified both sections for greater clarity**, now P16 and P7, to more clearly describe that cell spacing is not different when using soma centers, but that cell membranes are closer. The revised section on P16 now reads “Our empirical observations and mathematical modelling indicatethat for any average ChI or SPNs, their mean nearest neighbor astrocyte is at a similar, and predictable, soma center-soma center distance. We find however that astrocytes are more likely to form direct membrane apposition for with ChIs than for with SPNs, an observation that we speculate could arise from by the larger soma size diameters of ChIs.” We hope that more clearly explains the observations.

11. Why are changes in firing rate reframed as z-scores? what do you gain or what problem is it solving?

> Given that ChIs showed a spontaneous firing rate between 0 and 10 Hz, we thought representation of firing rate changes as z-score deviations from the mean would lead to more accessible and direct comparisons across experiments (e.g. drugs, opto/puff applications).

Reviewer #4

The manuscript by Stedehouder and colleagues is a thorough study of the effect of astrocytes activation on striatal dopamine release. In a series of experiments the authors show that astrocytes exert modulatory control over striatal cholinergic interneurons, thus modulating the activity of both DA axons and spiny projection neurons. Overall, the manuscript is well written, the experimental design is solid, and the findings are both novel and significant. I have some comments/suggestions that are listed below.

> We thank Reviewer 4 for their kind words, comments and suggestions.

It would be important to investigate the effect of astrocyte depolarization on striatal DA release during basic electrical stimulation of the slice. Can the authors show the effect of DREADD-mediated inactivation of astrocytes on DA release evoked by electrical stimulation?

> This is an interesting question, but previous attempts to use DREADDs to inhibit astrocytes identified that chemogenetic manipulations only activate, and not inactivate, striatal astrocytes: Stimulation of either Gq, Gs, or Gi pathways (with hM3D, rM3D, or hM4D DREADDs respectively) all activated astrocytic Ca^{2+} signalling (Chai et al 2017 Neuron 95, 531-49). Therefore, we did not invest resources in trialling DREADDs in this way. However, we have previously performed fluorocitrate-mediated inactivation of astrocytes, which strongly reduced electrical stimulation-evoked DA release (Roberts et al 2020, Nature Communications, PMID: 33009395).

Fig. 4. It would help if spontaneous AP frequency in excited and inhibited ChIs (for experiments shown on both a and l panels) is also shown since changes in voltage sag and AHP are expected to alter spontaneous neuronal firing. Same for the cell capacitance as a proxy of cell morphology. Additionally, if HCN channels and AHP components are important, the authors should measure I_h and AHP current in the voltage clamp mode, which gives a more accurate assessment of the real channel activity.

> We have now **added a new panel to Main Figure 4f** to depict spontaneous neuronal firing for excited and inhibited ChIs, which showed no overt differences between groups. We have not performed voltage clamp experiments to examine HCN/ I_h currents, although preliminary pilot data with the I_h blocker ZD7288 did not modulate astrocyte-induced ChI depolarizations.

Page 12. "These data indicate that rapid astrocytic modulation of pSPNs is indirect via a neuronal source of GABA release, which might include GABA-releasing ChIs." The statement is speculative without further experiments, including a demonstration of the effect of GABA_A (and potentially AChR) blockade on pSPN response to optogenetic stimulation of ChI (as on Suppl Fig 9r).

> We thank the reviewer for this perspective. To limit a sense that our statement might be over-speculative we **have rephrased this sentence** on P12 to be more cautious. That section now reads "These data indicate that rapid astrocytic modulation of pSPNs **seems** indirect via a neuronal source of GABA release, which **could potentially** include GABA-releasing ChIs^{43,45}. This mechanism was not the focus of interest here and was not explored further."

MINOR:

Fig. 4 Legend: “Excited ChIs compared to inhibited ChIs showed (d) a trend to lower input resistance compared to inhibited ChIs...” Should read ‘a trend to higher input resistance’.

> Thank you, **corrected**.

Suppl Fig S2. SR101 staining needs to be added to the Methods.

> We apologise for the omission, and have now **added!**

Suppl Fig S5a&h. Shouldn’t the X-axes read “Distance to the nearest astrocyte”?

> Thank you, **corrected**.

Suppl Fig S11i-l legend. “Z-scored firing rate responses in an example excitation ChI under 500-ms blue light activation” should read ‘inhibition ChI’.

> Thank you, **corrected**.

Fig 4 legend title needs some editing.

> Indeed! Thanks for spotting. Now **edited**.

Page 12. “The TTX-sensitive depolarization of pSPNs by astrocytes (see Supplementary Fig. 6)...” This should refer to Supplementary Fig. 9.

> Thank you for spotting. **Corrected**

Reviewer #5

> We thank Reviewer 5 for their time in reviewing our study, and helping to improve the quality of this manuscript.

Stedehouder et al NCOMMS-24-03524

REVIEWER COMMENTS and RESPONSES

> We thank all the reviewers for their helpful comments, which we have addressed below and through inclusion of some additional new data.

Reviewer #1

This is a very nice study. I don't have any additional comments and think the authors have done an excellent job of addressing all the Reviewer comments from all four Reviewers.

> Thank you.

Reviewer #2

The authors have made some changes in the manuscript to downtone the idea that astrocytes are electrically active and they have shown that neuronal ACh is released in the NAcc in response to astrocyte optoactivation. Moreover, they demonstrate that astrocyte fast depolarizations occur in response to glycine, which is a fantastic addition to the manuscript. Unfortunately, they have not satisfactorily addressed 3 of my previous concerns:

1. The fact that Ca²⁺ influx through ChR2 can be happening.

The authors state that “only a small fraction of ChR2 current is carried by Ca²⁺ influx, as we cited on original P17, and therefore we deem this an unlikely contribution”. However, the paper cited (Lin et al Bioph J), as well as Nagel et al PNAS, show that ChR2 indeed permeates Ca²⁺. Thus, it is possible that the calcium clearance is due to calcium entry through ChR2. If this is the case, I still think that it is improbable that such a mechanism exist in physiology. To rule out this possibility, I still think that they should either try to reproduce the results by depolarizing patched astrocytes, show that cytosolic calcium is not increasing in astrocytes upon optostimulation (in the presence of thapsigargin to rule out calcium coming from internal stores), or try to block some of the channels/exchangers they mention in the discussion (Cav, NCX or NCKX18).

> We thank the reviewer for their further comments, and we are glad that glycine data was appreciated for showing fast small depolarizations of astrocytes in response to neurotransmitters, similar to those following optogenetic stimulation. The specific issue of Ca²⁺ entry mediated via ChR2 is one that cannot easily be resolved; we acknowledge it as a very important consideration, and it is an issue for the field at large, relevant to more than just our manuscript. We have considered it here alongside the many factors at play. The reviewer has suggested some experiments to rule out Ca²⁺ entry via ChR2, or to validate Ca²⁺ entry via other channels. However, these experiments will be challenging and unlikely to be definitive. Channel blockers will not be selective for astrocytes but will impact on neurons also. Experiments to rule out changes to cytosolic Ca²⁺ in response to optogenetic activation are likely to be confounded by the hypothesis that Ca²⁺ entry (through ion channels) is likely to be central to the mechanisms at play. Given that we see a decrease in extracellular Ca²⁺, and that astrocytes use dynamic intracellular Ca²⁺ levels as a signalling modality, we think it likely that Ca²⁺ entry to astrocytes is occurring via potentially diverse Ca²⁺ channels on astrocytes, and that optogenetic activation will therefore modify cytosolic Ca²⁺ levels, whether or not Ca²⁺ entry is a component of the current carried by ChR2. **We have checked and refined our discussion section to ensure it (P17-18) covers all these possibilities, and have added a more explicit statement that Ca²⁺ conductance of ChR2 might also contribute.** We hope that this explanation and edits help to make sense of our thinking, alongside our glycine data.

2. Key experiments are missing in the NAcc

Although similar in cellular composition, the DLS and the NAc are distinct brain regions with differences in their inputs and outputs, dopaminergic innervation and involvement in behavior. Thus, one should not assume that the results found in the DLS will also apply in the NAc, and vice versa. I understand that repeating all experiments in both brain regions is redundant and it is not necessary. However, key experiments should be done to show that the same mechanism applies in both brain regions.

Since most of the work has been performed in the NAcc, they should demonstrate the involvement of Ch1 in dopamine release regulation. The observation that neuronal ACh is released upon astrocyte optoactivation is a good addition to the manuscript but it is not a causal demonstration for Ch1 involvement in dopamine release.

> To address this request, we have now **included the additional companion data for NAcc in a new figure panel (new Supplementary Fig S3h)**, in which we show that, as seen for DLS (Fig. 1m), the optogenetic activation of astrocytes in NAcc does not modify electrically evoked DA release when the nAChR antagonist DH β E is present.

If the authors are not willing to perform more experiments in the DLS, they should state in the discussion that a different mechanism may exist in this brain region.

> To address this point regarding the mechanisms operating in DLS to which we believe the reviewer is referring we have **added new text** to the Discussion at two separate junctures on P17 to specify that the mechanisms uncovered are those identified specifically “*in NAcc*”, and on P18 have added a statement that “*We did not explore the mechanisms operating in DLS.*”

3. Can extracellular Ca²⁺ changes also regulate dopamine release?

The authors have not provided evidence for the role of extracellular Ca²⁺ on dopamine release. They have shown that astrocyte optoactivation can regulate dopamine release in both the NAcc and the DLS, in the DLS this requires Ch1 activation. However, the involvement of Ch1 in dopamine regulation in the NAcc has not been demonstrated in this manuscript.

On the other hand, they show that astrocyte optoactivation regulate the activity of Ch1 in the NAcc by decreasing extracellular Ca²⁺ levels. Therefore, it is not directly explored whether the regulation of dopamine release is due to a decrease in extracellular Ca²⁺ levels or through a totally different mechanism.

> We believe that we have now addressed the point about involvement of Ch1s in dopamine regulation in NAcc with the new data included for point 2 above, which shows that optogenetic activation of astrocytes in NAcc does not modify electrically evoked DA release when in the presence of nAChR antagonist DH β E (**new panel Suppl Fig S3h**).

In addition, in relation to Ca²⁺, although dopamine release like all exocytosis is strongly Ca²⁺-dependent, when nAChRs are antagonised there are minimal effects of astrocyte activation on dopamine release in DLS or NAc (Fig 1m or S3h), indicating that the modulation of extracellular Ca²⁺ by astrocytes is not sufficient to strongly influence the level of DA exocytosis. It is modest and/or local and something to which Ch1s are highly sensitive.

Reviewer #2 (Remarks on code availability):

I am sorry but reviewing code is beyond my skills.

The code for models 2 and 3 are available in open access in Zenodo as a single word document. They state that the code can be run in python although I have not tried it.

Reviewer #3

As I pointed out in the first round of reviews the manuscript provides intriguing new observations regarding astrocytic control of cholinergic interneurons' control of dopamine release.

The reviewers suggested several reasonable requests for experiments in order to clarify either the mechanism, the physiological relevance or the interpretation of the findings. The authors have turned down many of the suggested experiments. For a journal like Nature Communications the authors should (and the reviewers do) expect additional experiments and leave to a bare minimum the experiments that are to be refused. I therefore think that the author need to conduct at the bare minimum the following two experiments that were suggested in the first round of reviews before I can consider the manuscript further.

I think reviewer 2's request to test whether depolarizing patched astrocytes can lead to a similar effect as that observed with Chr2 is a pertinent and easily achievable given that the authors are able to patch these cells. Particularly, since a strong depolarization of a single neuron by tens of millivolts should simultaneously depolarize many other astrocytes in the syncytium by a few millivolt so they should get the same result. If the experiments fails there is no harm, but if it works it will be a much stronger and physiologically relevant result if a localized cluster of astrocytes can generate the same effect

> We have made significant efforts to address this reviewer's comment, by **trailing double patching** of an astrocyte and a nearby ChI under dual visualisation (of Cre-dependent mCherry in ChIs, and GFAP-driven eGFP in astrocytes). We have attempted to briefly depolarise a local astrocyte by a few mV (4 mV) (to mimic the response to optogenetic or glycine-evoked depolarization) while monitoring firing rates of a local ChI. These experiments are non-trivial and low yield: dual patch is technically challenging in any event, and in addition, astrocytes outnumber ChIs by ~8 to 1 necessitating the expression and visualisation of complementary dual reporters to locate nearby pairs of each cell type. Furthermore, knowledge is very limited as to whether directly depolarizing the localised soma of a single patched striatal astrocyte will impact on activity in the astrocyte and branch network, or impact on adjacent astrocytes in any coupled syncytium. In our pilot experiments to date, we have not yet been able to see an impact of depolarisation of a single astrocyte on firing rate of single local ChI in these low yield experiments (see pilot data inserted below). We cannot ascertain from these experiments to date whether or not direct depolarisation of single astrocytes might be able to change ChI activity: there might be an insufficient level of activation of the patched astrocytes soma or branch network, or insufficient proximity of selected cell pairs, or it might be that activation of multiple astrocytes by a network of inputs is required. We have included our pilot data here, but feel it is too preliminary to include in the manuscript, while further development of this approach is currently beyond the scope of this project.

The statistical modeling and analysis of the authors regarding the privileged proximity of CINs to astrocytes is nice, but it's hard to judge its veracity. This is why I asked to conduct a similar analysis regarding the proximity of astrocytes to another class of GABAergic interneurons (GINs). Either way this experiment comes out would be valuable to the authors and the readers, and I don't see why the authors refuse to do it. If it turns out that astrocytes are as close to GINs that could explain the polysynaptic inhibition of CINs (if astrocytes can also directly excite GINs). It is really critical to find out whether this mechanism is unique to CINs or more general, so I think that addressing the possibility that GINs have a privileged status is critical.

> As requested, we have now **conducted an additional analysis** for proximity of an example class of GABA interneurons (PV-positive GINs) to astrocytes. We analysed specifically the occurrence of soma-to-soma membrane appositions, as it is this feature that distinguished ChIs from SPNs. We found only a low 5-10% occurrence of soma-to-soma membrane appositions between PV-interneurons and astrocytes, similar to the ~8% value we report between SPNs and astrocytes, and less than the 20-43% occurrence seen between ChIs and astrocytes. We have **included this additional new result for PV interneurons in the Results (P7)**, and illustrated successful PV immunolabelling **in new panel Suppl. Fig S5h**, and we have updated the relevant sections of Methods and discussion accordingly. These new data strengthen further the findings of our manuscript, pointing to a relatively privileged interaction between astrocytes and ChIs.

In summary, because the phenomena are interesting but also very variegated, conducting more experiments is warranted to help clarify what is going on.

> We hope that all of our additional new data and information clarifies further for the reviewer.

Reviewer #4

The authors addressed all of my questions/concerns and I recommend the manuscript for publication.

> Thank you!

Reviewer #5

> Thank you for your support of the peer review process.

Stedehouder et al NCOMMS-24-03524B

REVIEWER COMMENTS and RESPONSES

> We again thank all the reviewers for all their time and input, which have helped to improve our manuscript.

Reviewer #2

The authors have now performed experiments in the NAc providing a more complete story in this brain region, which may similarly occur the DLS.

On the other hand, once again, the authors have overlooked my main criticism. The whole idea that astrocytes regulate neuronal firing by buffering calcium is challenged by the fact that ChR2 can permeate calcium. I can still argue that what they have observed is simply an artifact. In my opinion, one can speculate what is the mechanism of a phenomenon when multiple possibilities exist. However, if one of those possibilities is that the findings could be an artifact of the chosen method, it becomes essential to perform proper controls.

If the authors believe that calcium is entering through channels, blocking those channels with a cocktail of antagonists and assessing astrocyte calcium while activating them with ChR2 is such an easy experiment that I genuinely cannot understand why they refuse to do it. It even makes me think that they have tried and the experiment didn't work. Notably, according to the response to Reviewer 3, the authors have attempted one of the experiments I suggested to address this issue: depolarizing astrocytes and monitoring firing activity in ChI. However, these experiments have not yet been successful.

The authors argue that this point is relevant for the field and not only for their manuscript. I fully agree, but, just because other people do it doesn't mean it is correct. In fact, other papers using ChR2 in astrocytes combine this technique with other complementary approaches or control experiments to demonstrate that the observed effects are not artifacts, unfortunately this is not always the case.

> We thank the Reviewer for their continued input. We have not overlooked this point. We have not attempted to untangle conductances operating through ChR2 on astrocytes because any such experiments would be extremely challenging, and data would be unlikely to be definitive. To light-activate ChR2 in astrocytes and simultaneously image intracellular calcium would itself be a challenge, one we are not set up to perform. And if we attempt to block candidate Ca^{2+} channels, neither negative nor positive outcomes would be definitive. A lack of detectable impact on Ca^{2+} levels would always leave open the explanation that we have simply not blocked the appropriate route of entry, given the limitation in knowledge or tools for all the plausible routes of Ca^{2+} entry to astrocytes, making complete block impossible. Conversely, a strong impact that appeared to define the route of entry, could alternatively be attributed to non-selective drug effects on neurons in interacting circuits that shape astrocytic signalling or simply to a level below reporter signal threshold. Evidence would at best be circumstantial. Rather, to address this point we conducted complementary experiments described in an earlier revision, in which we showed that depolarization in astrocytes is also an outcome of input from striatal neurotransmitter input, using glycine as an example. Nonetheless, we continue to acknowledge the issue, and have again **revised the corresponding section of the Discussion with a revised sentence to more strongly flag that this issue remains at play**. The sentence on P18 now reads "However, the modest Ca^{2+} conductance of ChR2 might also contribute⁷⁴ and this cannot easily be excluded."

Reviewer #3

I appreciate the additional experiments on PV interneurons, which strengthen the manuscript. I have no more concerns. Congratulations on an intriguing set of findings.

> Thank you!